American Society for Microbiology | Microbiology Spectrum

# A Deep Learning Approach to Capture the Essence of *Candida albicans* Morphologies

Van Bettauer,[a] Anna Carolina Borges Pereira Costa,[b] Raha Parvizi Omran,[b] Samira Massahi,[b] Eftyhios Kirbizakis,[b] Shawn Simpson,[a] Vanessa Dumeaux,[c] Chris Law,[d] Malcolm Whiteway,[b] Michael T. Hallett[b,e]

[a]Department of Computer Science and Software Engineering, Concordia University, Montreal, Quebec, Canada
[b]Department of Biology, Concordia University, Montreal, Quebec, Canada
[c]Department of Anatomy and Cell Biology, Western University, London, Ontario, Canada
[d]Centre for Microscopy and Cellular Imaging, Concordia University, Montreal, Quebec, Canada
[e]Department of Biochemistry, Western University, London, Ontario, Canada

**ABSTRACT** We present deep learning-based approaches for exploring the complex array of morphologies exhibited by the opportunistic human pathogen *Candida albicans*. Our system, entitled Candescence, automatically detects *C. albicans* cells from differential image contrast microscopy and labels each detected cell with one of nine morphologies. This ranges from yeast white and opaque forms to hyphal and pseudohyphal filamentous morphologies. The software is based upon a fully convolutional one-stage (FCOS) object detector, a deep learning technique that uses an extensive set of images that we manually annotated with the location and morphology of each cell. We developed a novel cumulative curriculum-based learning strategy that stratifies our images by difficulty from simple yeast forms to complex filamentous architectures. Candescence achieves very good performance (~85% recall; 81% precision) on this difficult learning set, where some images contain hundreds of cells with substantial intermixing between the predicted classes. To capture the essence of each *C. albicans* morphology and how they intermix, we used a second technique from deep learning entitled generative adversarial networks. The resultant models allow us to identify and explore technical variables, developmental trajectories, and morphological switches. Importantly, the model allows us to quantitatively capture morphological plasticity observed with genetically modified strains or strains grown in different media and environments. We envision Candescence as a community meeting point for quantitative explorations of *C. albicans* morphology.

**IMPORTANCE** The fungus *Candida albicans* can "shape shift" between 12 morphologies in response to environmental variables. The cytoprotective capacity provided by this polymorphism makes *C. albicans* a formidable pathogen to treat clinically. Microscopy images of *C. albicans* colonies can contain hundreds of cells in different morphological states. Manual annotation of images can be difficult, especially as a result of densely packed and filamentous colonies and of technical artifacts from the microscopy itself. Manual annotation is inherently subjective, depending on the experience and opinion of annotators. Here, we built a deep learning approach entitled Candescence to parse images in an automated, quantitative, and objective fashion: each cell in an image is located and labeled with its morphology. Candescence effectively replaces simple rules based on visual phenotypes (size, shape, and shading) with neural circuitry capable of capturing subtle but salient features in images that may be too complex for human annotators.

**KEYWORDS** *Candida albicans*, deep learning, fully convolutional one-stage object detection, microscopy, morphology, generative adversarial network

Address correspondence to Michael T. Hallett, michael.hallett@uwo.ca.
The authors declare no conflict of interest.

Fungal infections represent an urgent and significant threat to human health affecting 1.2 billion people yearly (1–3). They kill approximately the same number of people (1.6 million) as malaria (4) and are implicated in other diseases, including cancer progression (5). *Candida albicans* is one of the most important of these human pathogens (6) and is associated with a significant socioeconomic burden (2, 7–9). As such, it is an important system for studying fungal pathogenicity. *C. albicans* is morphologically classified as a pleomorphic yeast-like fungus, and its diverse range of morphologies predominate in different niches. The morphologies can be broadly partitioned into mating-competent and vegetative forms (10, 11). We provide a brief review of the morphologies central to our effort here (Fig. 1A to E).

Mating-competent morphologies include the white form, which has a round unicellular morphology (Fig. 1A and B), predominantly found in the bloodstream and internal environments (12). If engulfed by macrophages, the white yeast will switch to a hyphal morphology as a means of immune escape (13). The opaque form is larger and more elongated than the spherical white cells; it often colonizes skin (14). Switching between the two morphologies is rare ($\sim 10^4$ cell divisions), stochastic, and strongly influenced by environmental cues and homozygosity of mating type. White cells are believed to be better suited to internal infections, while opaque cells thrive in skin infections (14). The gray morphology is an alternative, stable form between the white and opaque morphologies. Gray cells are shiny and small like white cells yet elongated like opaque cells in solid media (15). The shmoo morphology arises from opaque cells which have formed a projection produced by the **a** and $\alpha$ types when preparing to mate, leading to tetraploid zygote formation.

*C. albicans* can also assume vegetative morphologies, including the budding yeast white cell and two distinct filamentous forms (Fig. 1C to E). The hyphal form is characterized by long tube-like filaments without constrictions at the site of septation. Hyphae are able to invade epithelial and endothelial cells and damage host tissue in mucosal infections in order to gain access to the bloodstream (16) (Fig. 1D). The yeast-to-hypha switch is initiated under a variety of environmental conditions, including presence of serum, neutral pH, 5% $CO_2$, and *N*-acetyl-D-glucosamine, among others (17). In the first cell cycle, the germ tube morphology is observed, manifesting as a tube projecting from the round yeast cell. The second filamentous growth form, the pseudohypha, is significantly different from hyphae in many ways, including cellular regulation, growth (18–20), and phenotypic presentation. Pseudohyphae tend to have constrictions at the sites of septation and are wider than hyphae (Fig. 1E) (17). Cell divisions in pseudohyphae are typically nearly synchronous, unlike hyphae, where subapical cells are often arrested in $G_1$ for several cycles (21). Like hyphal cells, pseudohyphae interact with the mouth, vagina, and bloodstream of the host. Some *C. albicans* morphologies are not considered in our effort here, including the chlamydospore, gastrointestinally induced transition (GUT), trimera, and goliath morphologies (22, 23).

There have been several previous efforts in fungal image analysis (24), including approaches to computationally differentiate between types of fungi and allergenic fungal spores (25) and to characterize the macrostructure of mycelium or filamentous growth (26–33). Tleis and Verbeek experimented with a suite of machine learning techniques to segment *Saccharomyces cerevisiae* cells and measured a range of features and textures from two-channel images acquired by laser scanning confocal microscopy (34). Wang et al. provided a segmentation method with an efficient edge-tracing algorithm for bright-field images of fission yeast, which have a consistent oblong morphology (35). Another effort employed microfluidics to capture individual *S. cerevisiae* cells; nonfluorescent images were used to train a classifier of cell cycle state for each cell (36).

Deep learning, a toolkit exploited in our effort here, was first applied to biomedical imaging in 2012 with the approach by Ciresan and colleagues (37) to automatically segment neuronal structures depicted in stacks of electron microscopy images. Several image segmentation approaches appeared thereafter, including U-net from Ronneberger et al. (38) and a now well-established deep learning architecture entitled Res-Net (39),

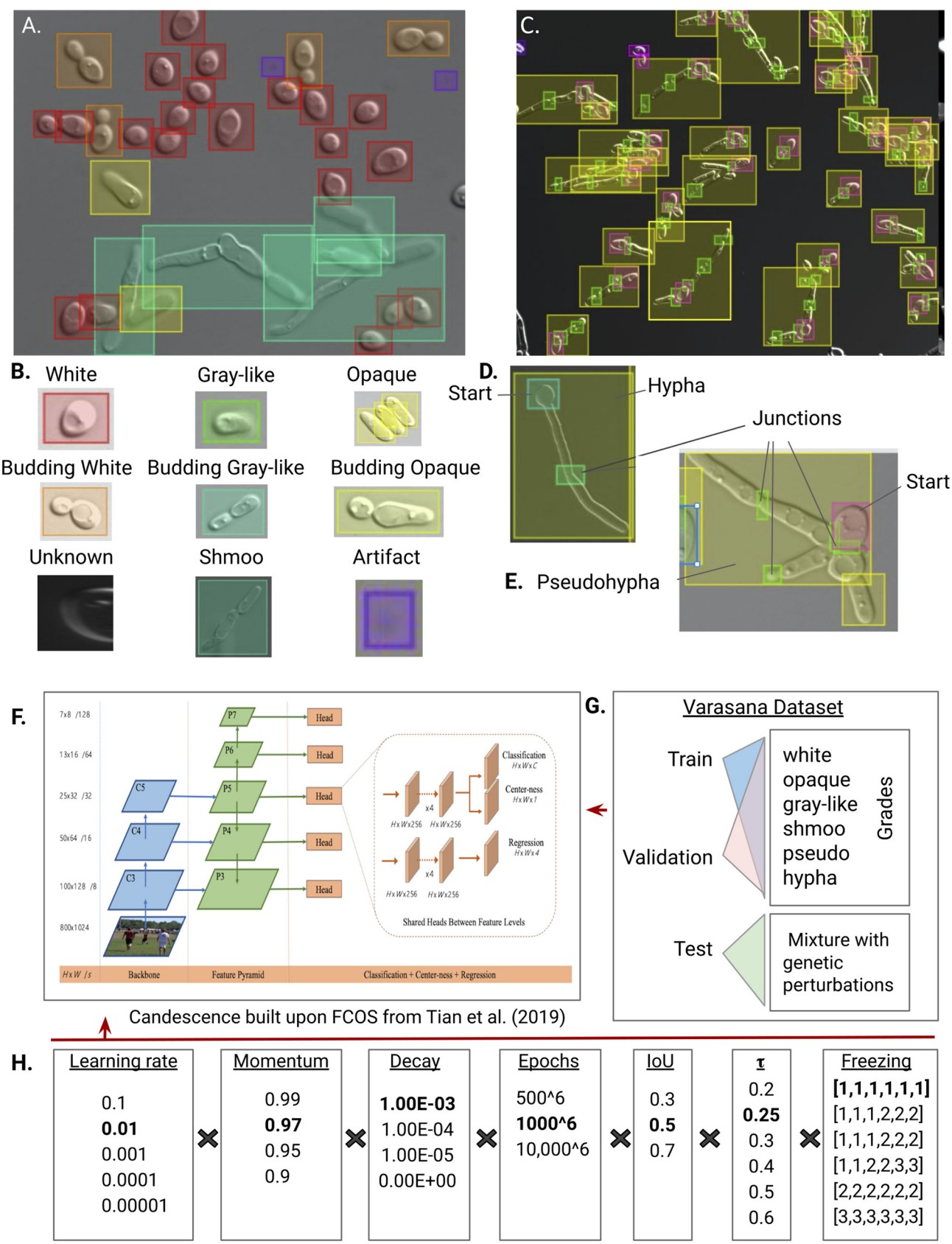

**FIG 1** (A) Example of a typical DIC image in Labelbox, a learning set creation tool for group annotation. (B) The colors indicate class of morphology. The "artifact" class is used to bound imperfections and technical artifacts in the images, whereas the "unknown" class is used to bound cells for

which we exploited here. These have been extended to microscopy images, although they are primarily used in the context of human tissue analyses (40–43). Nevertheless, deep learning has introduced many new concepts, transformations, object localization techniques, and strategies (44–47). This includes unsupervised generative adversarial training (48) and variational autoencoder-based methods (49) and supervised methods such as the faster region-based convolutional neural network (Faster R-CNN) (50). This effort exploits both supervised and unsupervised deep learning.

With respect to fungi, there has been a significant effort to integrate robotically controlled hardware with high-content image analysis tools to rapidly screen and analyze hundreds of thousands of images of *S. cerevisiae* in order to identify essential genes and genetic interactions (51, 52). These works were the first to bring large-scale image analysis at the subcellular level, including DeepLoc, a deep learning approach to *S. cerevisiae* (53). With respect to *C. albicans*, Merson-Davies and Odds in 1989 were the first to develop a quantitative model for differentiating between hyphae and pseudo-hyphae based on the measurements of, for example, width of the tubes from microscopy images (54). Shortly thereafter, this group used these models to reason about the nature of morphological plasticity (55).

Our effort here extends the power and type of models to identify and capture *C. albicans* morphology by exploiting modern techniques from deep learning. We first developed a system that can automatically detect *C. albicans* cells from microscopy images and label each detected object with its morphology. The model was trained using our substantial compendium of differential interference contrast (DIC) images ($\sim$1,200 images) containing individual *C. albicans* cells ($\sim$80,000) in different morphological states. We treated this problem as a multiclass, multiobject object detection problem where 15 distinct classes are used to describe nine morphologies. Some images contain as many as two hundred tightly packed cells. Our method builds upon a fully convolutional one-stage object detector (FCOS) (56); FCOSs are more computationally efficient than techniques such as Faster R-CNN (they are proposal and anchor free) and better predict the location of objects (via the "centerness" optimization). We also designed a novel type of curriculum-based learning strategy that stratifies images by difficulty but uses a cumulative approach that improved the performance of our system significantly. Then, building upon our ability to accurately identify and classify cells, we developed a deep learning-based model that captures the essence of each *C. albicans* morphology. This model, which is based upon a generative adversarial network (GAN), can be interrogated to identify components of its latent space that control various features of the images. This includes technical variables but also biologically relevant processes such as developmental trajectories or transitions between morphologies. These models provide the first dynamic and continuous approach to capturing the essence of the different morphological states and transitions between them. We show how these models can be used to automatically identify subtle changes from wild-type phenotype when given images from genetically perturbed *C. albicans* populations.

## RESULTS

**A deep learning approach to recognizing *C. albicans* morphologies.** Our first goal was to develop a fully automated tool that identifies the location and morphology of cells from microscopy images of *C. albicans* populations. We cast this challenge as a multiobject, multiclass detection problem: the system (i) identifies the location of all

**FIG 1** Legend (Continued)

which we could not judge morphology. (C) Labeling of an image that is enriched for pseudohyphae. Note that the overall intensities of our images are not required to be the same. (D) Given the size and complexity of the filamentous forms, we annotated each hypha or pseudohypha using three classes. (D and E) H-start and P-start are intended to label only the "start" of the hypha or pseudohypha. Since junctions are different between hyphae and pseudohyphae, we labeled them with the H- and P-junction classes. Finally we bounded the entire filamentous object with a rectangle; note that such bounding boxes overlap surrounding cells. (F) Our system is built upon the FCOS software with structure unaltered from Tian et al. (56). (G) The Varasana learning set is first partitioned into three components (training, validation, and testing). Then both validation and testing are split into six grades. The grades are ordered from white to hypha. When an image appears first in grade $i$, it is also included in all grades $j > i$. The test set contains examples of both wild-type SC5314 and SN148a cells but also a collection of genetic and environmental perturbations that induced abnormal *C. albicans* morphologies. (H) Using the Varasana learning set, we performed a grid search across several hyperparameters ($n = \sim$30,000 combinations). Boldface indicates the combination of hyperparameters that had the best performance.

*C. albicans* cells in each image and then (ii) predicts the morphology of each hypothesized cell. Images may contain an arbitrary number of individual objects (cells or artifacts). Our system classifies nine of the 12 reported *C. albicans* morphologies: yeast white, budding yeast, opaque, budding opaque, shmoo, gray-like, budding gray-like, hypha, and pseudohypha. Each detected object is assigned the morphology with the highest likelihood, provided that the probability of the prediction exceeds a user-defined threshold, $\tau$.

Our software (entitled Candescence) is based on a FCOS (56), a new approach that has several benefits over other deep learning algorithms for multiobject, multiclass problems (Fig. 1F). One of its key advantages resides in how it flags areas of an image likely to contain an object. Rather than requiring many parameters, which collectively control the size, location, and total number of bounding boxes, the FCOS considers each individual pixel in an image as potentially centering an object, eliminating the need to optimize many hyperparameters simultaneously. FCOSs are able to handle objects of variable size, an important property given the difference between, for example, yeast white and hyphal cells.

FCOSs can make use of transfer learning, a technique where a neural network is first trained in a distinct but similar context compared to the problem at hand. In our case, we began with an FCOS trained with the ResNet-101 data set, a convolutional neural network trained on more than a million images partitioned into 1,000 classes of common household objects and animals (39). Intuitively, transfer learning guarantees that our neural network comes equipped with the basic circuitry to recognize simple shapes, shades, edges, and textures. This is especially important when the number of images for the new application is limited. Candescence exploits this rich feature representation to be retrained for our morphologies, requiring only a tractable number of *C. albicans* images. The architecture and technical parameters of the FCOS are described in "A fully convolutional one-stage object detector for morphology classification" in Materials and Methods.

**An image compendium of *C. albicans* morphologies.** Supervised machine learning problems, including this image recognition problem, exploit a learning set, which is a curated collection of images where bounding boxes (a box that surrounds the object) have been drawn and labeled for every object. We constructed such a learning set by growing *C. albicans* SC5314 or SN148a colonies under conditions which induce each of the nine morphologies (Table 1; also, see "Strains and media" in Materials and Methods): yeast white, budding white, opaque, budding opaque, gray-like, budding gray-like, shmoo, hypha, and pseudohypha. Each population was stained with calcofluor white and prepared for DIC and fluorescence microscopy at various magnifications (objective magnifications of 40, 60, and 100×; see "Microscopy" in Materials and Methods). We also cultured and imaged *C. albicans* colonies that had specific genetic perturbations known to affect morphology. In total, 1,214 images were generated (see Table S1 in the supplemental material).

In each of the resultant DIC images, a bounding box was manually drawn around each *C. albicans* cell and labeled with its morphological class following the guidelines of Sudbery et al. (20), Whiteway and Bachewich (11), Tao et al. (15) and Noble et al. (22) (see "Image annotation and development of the learning set" in Materials and Methods). More specifically, cells with a round to oval morphology (4.9 $\mu$m by 6.8 $\mu$m) were labeled as white unless they were attached to a second smaller white cell with concomitant evidence of a bud site (decision making was assisted by manual inspection of the calcofluor white signal in the matched fluorescent image); in this case, they were labeled as budding white (Fig. 1A and B). Ellipsoidal cells approximately twice the size of white cells (9.5 $\mu$m by 11.8 $\mu$m) were labeled as opaque. Like white cells, they were labeled as budding opaque if there was evidence of a smaller opaque cell with a new bud site. Cells were labeled as shmoo if they had an irregular (often boomerang) shape and large vacuoles. Many of our images contain a significant number of mating-competent cells that are visibly distinct from both white and opaque morphologies.

**TABLE 1** Summary of the Varasana learning set from Table S1

| Use[a] | Type(s) | Genetic perturbation(s) | Serum[b] | Temp (°C)[c] | Predominant morphology | Significance[d] |
|---|---|---|---|---|---|---|
| A | 64 | SN148a | No | RM | White | NA |
| | 42 | SN148a | No | RM | Opaque | NA |
| | 41 | SN148a | No | RM | White and opaque | — |
| | 54 | SN148a | No | RM | Shmoo | — |
| | 55 | SN148a | No | RM | Shmoo and white | — |
| | 10 | SC5314 | No | 30 | White | — |
| | 11 | SC5314 | 20 | 37 | White | — |
| | 65 | SC5314 | No | 30 | Pseudohypha and hypha | — |
| | 66 | SC5314 | No | 37 | Pseudohypha and hypha | — |
| | 1, 5, 6 | RHA1 GOF | No | 30 | Pseudohypha and hypha; small | — |
| B | 4 | RHA1 GOF | 10 | 37 | Pseudohypha | — |
| | 3 | RHA1 GOF | 20 | 37 | Pseudohypha | — |
| | 34 | UME6 null | No | 30 | White | — |
| | 27 | BRG1 null | No | 30 | White | — |
| | 21 | BRG1 null | 20 | 37 | Pseudohypha | * |
| | 8 | RHA1 null | No | 30 | White | — |
| | 9 | RHA1 null | 20 | 37 | Pseudohypha/hypha; low branches | * |
| | 7 | RHA1 null, RHA GOF | No | 30 | White | — |
| | 2 | RHA1 GOF, UME6 null | No | 30 | | ** |
| | 46 | RHA1 GOF, BRG1 null | 20 | 37 | | ** |
| | 16, 18, 19 | RHA1 GOF, BCR1 null | No | 30 | | ** |
| | 14 | RHA1 GOF, BCR1 null | No | 37 | | ** |
| | 13, 17, 20 | RHA1 GOF, BCR1 null | 20 | 37 | | — |
| | 15 | RHA1 GOF, BCR1 null | 10 | 37 | | — |
| | 36 | RHA1 null, BRG1 null | 10 | 30 | | ** |

[a]A, some images from this type of colony appeared in either the training or validation data sets; B, some images were used in the test set. All genetic perturbations were performed in SC5314 cells.
[b]"No" indicates that only YPD medium was used; values are the percent serum added to YPD.
[c]RM, room temperature.
[d]Results of applying the proportionality test to determine if there was a difference in performance between the validation and test set. *, $P < 0.05$; **, $P < 0.01$; —, $P > 0.05$. In cases where a $P$ value is provided for a type A colony, only images that were omitted from the training and validation data sets were used. NA, not applicable.

These cells are ellipsoidal, with a size in the upper quantile of white cells but below the size of opaque cells. These properties are close in spirit to the gray morphology of Tao and colleagues (15), although we did not grow *C. albicans* colonies under conditions that specifically induce this morphology. To date, confirmation of the gray cell morphology requires the use of specific molecular markers. However, given the distinctiveness and ubiquity of these cells in our data, we decided to explicitly model this structure and use the term "gray-like" to distinguish them from "true" gray cells according to Tao and colleagues.

The diversity and complexity of filamentous cells necessitated the development of an approach that uses several markers concomitantly to reliably identify them. Hyphae are thin tube-shaped cells with a width of ~2 $\mu$m. Pseudohyphae are multicellular entities which tend to be elongated and ellipsoidal. The minimum width is 2.8 $\mu$m. Our markup scheme first bounds the entire hypha or pseudohypha. Since filamentous cells are often long and irregular in shape, the bounding box almost always overlaps, or completely subsumes, the bounding box of other objects. Since this overlap represents a significant challenge for learning algorithms, we also labeled the location of the original germ bud of each hypha or estimated the start cell of the pseudohypha; these bounding boxes were labeled as H- and P-start, respectively. Compared to hypha/pseudohypha bounding boxes, H- and P-start are much smaller and therefore disjoint from other objects in the image (Fig. 1C to E). As is the case for the start sites, the septal junctions in hyphae and pseudohyphae are visually distinct from one another. Only pseudohyphae have constrictions at the mother-bud neck and subsequent septal junctions. We placed bounding boxes at these junctions and labeled them as H- or P-junction. Cells for which we could not reach agreement on morphology were labeled as unknown, and noncellular events in the images were labeled as artifacts.

We stress here that our strategy attempted to label each cell using only its visual appearance and independently of other factors, including the predominant morphology of cells in the image. For example, a colony grown under conditions which enrich for white cells is still expected to harbor a small number of opaque cells (1 in $10^4$); our procedure would label a cell as opaque in this case. However, our labeling was likely biased towards an assignment of white for a cell exhibiting both white and opaque attributes. In some cases, a consensus was difficult to achieve across labelers.

**Varasana: a cumulative curriculum-based *C. albicans* learning set for Candescence.** Learning sets are typically tripartitioned (Fig. 1G). The training set contains a sizeable set of images which are presented to the neural network during training and used to fit the model (i.e., update the weights of each arc in the neural network). The validation set is typically a smaller sample of data which is used to generate an unbiased estimation of the model fit during training. It provides a means to tune the hyperparameters of the model. The test data set is used only once after all training is complete to assess the final fit of the model. We partitioned the images referenced in Table 1 so that cells from each morphology are assigned to the training and validation sets at a 7:3 ratio. This procedure was confounded by the fact that there is significant variability in both the number of cells per image and the composition of morphologies per image. From the collection of 1,214 images, the training and validation data sets contain 216 and 94 images, respectively. These images in turn contain 4,880 and 1,958 objects, respectively. The independent test contains the remaining 904 images.

FCOSs have several hyperparameters that affect the overall performance of the system, including the learning rate (the amount weights are updated during gradient descent), momentum (a parameter that stipulates how many previous steps can be used to determine the direction of a weight update), decay (a regularization parameter restraining the complexity of our model), epoch number (the number of times the learning phase cycles through the complete training and validation sets), the IoU (the intersection of union statistic, which controls how accurate the regression of the bounding box must be), the threshold $\tau$ (the lower bound for the probability an object is assigned a class), and others (Fig. 1H). We performed a grid search across a range of values for these parameters and measured convergence, model complexity, and performance after each trial (see "Image annotation and development of the learning set" and "Measures of system performance" in Materials and Methods). Although the resultant classifiers had good performance for some morphologies (e.g., $F_1$ was ~78% for white, budding white, opaque, gray, and shmoo), several classes remained poorly predicted (e.g., $F_1$ was ~50% for pseudohypha- and hypha-related classes), and budding classes were poorly distinguished from their parent classes (e.g., $F_1$ was ~60% for budding white, gray, and opaque).

The steep increase in difficulty between yeast white and the filamentous classes highlighted the need for a more structured learning set. We opted for a curriculum approach (57, 58), a well-established concept in psychology which has re-emerged in the deep learning community. The fundamental idea is to structure the learning set so that the neural network is exposed to concepts according to their difficulty, with the easiest concepts presented first. Using the results from our original grid search to judge easy and difficult objects, we redesigned our learning set into grades 1 through 6. Each grade was enriched for a specific subset of classes, although at least one example of all classes was present at each grade. In general, grade 1 is enriched for yeast white and budding white cells, grade 2 introduces opaque and budding opaque, grade 3 presents gray and budding gray, grade 4 focuses on the shmoo form, grade 5 focuses on pseudohypha, and grade 6 concludes with hypha. Once an image appears at a certain grade, it appears in all subsequent grades. We hypothesize that this cumulative strategy ensures that lessons learned early with populous morphologies such as yeast white and opaque are retained when the complicated filamentous morphologies are presented to the learner. To the best of our knowledge, this is the first use of a cumulative approach with curriculum learning.

**TABLE 2** Exploration of the performance of the FCOS in the grid search (IoU = 0.5)[a]

| Parameter | Result for a $\tau$ value of: | | | | | | | | | |
|---|---|---|---|---|---|---|---|---|---|---|
| | 0.05 | 0.1 | 0.15 | 0.2 | 0.25[b] | 0.3 | 0.35 | 0.4 | 0.5 | 0.6 |
| Total number of ground truth objects | 2,387 | 2,479 | 2,526 | 2,552 | 2,563 | 2,564 | 2,568 | 2,572 | 2,575 | 2,575 |
| Blind spots | | | | | | | | | | |
| Number | 252 | 270 | 281 | 289 | 301 | 327 | 344 | 373 | 440 | 557 |
| Rate | 0.106 | 0.109 | 0.111 | 0.113 | 0.117 | 0.127 | 0.134 | 0.145 | 0.171 | 0.22 |
| Hallucinations | | | | | | | | | | |
| Number | 482 | 445 | 414 | 384 | 358 | 342 | 325 | 306 | 272 | 218 |
| Rate | 0.202 | 0.18 | 0.164 | 0.15 | 0.14 | 0.133 | 0.127 | 0.119 | 0.106 | 0.09 |
| Adjusted hallucinations | | | | | | | | | | |
| Number of true hallucinations | | | 8 | 10 | 13 | 14 | 17 | | | |
| Number misclassified | | | 22 | 26 | 30 | 33 | 37 | | | |
| Number correct | | | 384 | 348 | 315 | 295 | 271 | | | |
| Classifications | | | | | | | | | | |
| Number of errors | 389 | 387 | 384 | 378 | 352 | 332 | 319 | 299 | 261 | 215 |
| Number correct | 1,338 | 1,397 | 1,415 | 1,416 | 1,407 | 1,386 | 1,368 | 1,330 | 1,277 | 1,270 |
| Rate | 0.775 | 0.783 | 0.787 | 0.789 | **0.8** | 0.807 | 0.811 | | | |
| Adjusted classifications | | | | | | | | | | |
| Number of errors | | | 406 | 404 | 382 | 365 | 356 | | | |
| Number correct | | | 1,799 | 1,764 | 1,722 | 1,681 | 1,639 | | | |
| Adjusted rate | | | 0.816 | 0.814 | <u>**0.818**</u> | 0.822 | 0.822 | | | |
| Sensitivity/recall | 0.842 | 0.838 | 0.834 | 0.83 | **0.824** | 0.809 | 0.799 | 0.781 | 0.736 | 0.65 |
| Adjusted sensitivity/recall | | | 0.865 | 0.859 | <u>**0.851**</u> | 0.837 | 0.827 | | | |
| Precision | 0.606 | 0.627 | 0.639 | 0.65 | **0.665** | 0.673 | 0.68 | 0.687 | 0.697 | 0.71 |
| Adjusted precision | | | 0.808 | 0.804 | <u>**0.807**</u> | 0.809 | 0.807 | | | |
| $F_1$ | 0.704 | 0.717 | 0.724 | 0.729 | **0.737** | 0.735 | 0.735 | 0.731 | 0.716 | 0.68 |
| Adjusted $F_1$ | | | 0.838 | 0.834 | <u>**0.832**</u> | 0.826 | 0.821 | | | |

[a]Relevant summary statistics for false negatives (blind spots), false-positive object predictions (hallucinations), and classifications. With *a posterior* analysis and correction of the hallucinations, performance was recalculated and reported in the adjusted sensitivity row. True positives correspond to a correctly identified object that is also correctly classified. In an FCOS, all pixels which are not part of a bounding box and which are not predicted to be part of a bounding box are true negatives. False positives are hallucinations and incorrect classifications. False negatives are blind spots only.
[b]Boldface indicates estimates of performance for the optimal value of $\tau = 0.25$; underlining indicates the final performance after adjustments.

A new grid search was conducted, but this time, the hyperparameters were allowed to vary across the different grades. This grid search also considered different levels of freezing in our four-layer FCOS. Freezing refers to the process of not allowing layers of the neural network to change. For example, the most restrictive freezing regimen does not allow any of the four layers to change in response to new examples, implying that our classification is solely based on the original transferred ResNet-101. The most permissive freezing strategy allows all layers to be updated during training when presented with our images. Figure S1 breaks down the learning set by grade and classes and provides the distribution of the number of cells per image.

**Searching for classifiers of *C. albicans* morphologies.** The grid search, which required ~60 days of continuous computation on a 10-GPU (graphics processing unit) system, was performed with the validation data set across the hyperparameter space. We judged performance initially using the mean average precision (mAP) (59), a standard approach for measuring the performance of multiobject/multiclass problems, across all trials with a final value of 0.407 (see "Measures of system performance" in Materials and Methods). This best classifier has a learning rate of 0.01, momentum of 0.97, and decay of 0.001 (Fig. 1H). Analysis of the loss curves established that 1,000 epochs at each grade sufficed for convergence for each of the three component loss curves (see "Measures of system performance" in Materials and Methods; also, see Fig. S2). An initial performance assessment suggested that an IoU of 0.5 and a $\tau$ of 0.25 maximized the recall, precision, and $F_1$, which were estimated to be 82.4%, 66.5% and 73.7%, respectively (Table 2).

Somewhat surprisingly, the performance of classifiers with more liberal freezing strategies was better than that with strategies that froze layers. We hypothesize that the cumulative nature of our learning set guaranteed that the network retained learned rules from early grades without the need for freezing. Figure 2 depicts the ground truth labels (left) and predicted objects and their classifications (right) across three typical images containing many of the morphologies. The images help to show that Candescence is able to cope with overlapping objects in dense images.

**Candescence rarely hallucinates (false-positive objects) but has blind spots (missed objects).** The performance of our classifier can be decomposed into its object detection and object classification components. For object detection, false positives refer to cases where Candescence predicts a bounding box in a location of the image that does not have a ground truth bounding box. Using the 94 images of the validation data set with an IoU of 0.5 and a range of thresholds for $\tau$, we manually examined all false-positive predictions. These are depicted in Fig. S3 and S4 (358 false object detections for a $\tau$ value of 0.25). It is difficult in some dark images ($n = 13$) to detect an object. We confirm these as false-positive object detections. In approximately 30 cases, there is in fact an object in the bounding box but the predicted classification is incorrect. Essentially, both the human labelers and Candescence have made mistakes of missing the bounding box and incorrectly assigning class, respectively. In all remaining 315 cases, Candescence was correct to predict an object that the human labelers had missed. After adjusting the performance measures by removing the effects of human errors, the recall, precision, and $F_1$ rose to 85.1%, 80.7% and 83.2%, respectively (Table 2, values in bold and underlined; also, see "Measures of system performance" in Materials and Methods). Figure 3A provides a small sample of correct and incorrect false positives, or "hallucinations."

Candescence, however, did miss several bounding boxes in the validation data set (301 false negatives). Approximately one-quarter of such blind spots correspond to the artifact class. Artifact is a heterogeneous class that was used to label all defects in the microscopy images regardless of their individual visual qualities. As such, it is perhaps understandable that Candescence did not learn to predict this class well, as there is no consistent set of attributes. Approximately one-quarter of the blind spots are related to objects labeled as yeast white in the ground truth data set (Table 3). We observed a recurring pattern across these cases: often, the white cell was physically adjacent to a second cell (Fig. 3B, panel i). Forensic investigation of the neural network suggests that the softmax function of the FCOS distributes the probability between the yeast white and budding yeast white uniformly. This caused the score for both categories to fall below our chosen threshold ($\tau$) of 0.25; hence, Candescence failed to identify a bounding box in that location of the image. This also occurred between opaque and budding opaque. In fact, more than one-third of the images in the validation data set had at least one problematic prediction of this form.

Candescence failed to identify several white cells in dense images (Fig. 3B, panel ii) and failed to label several germ tube cells as H-start. As well, P-junctions in pseudohypha-enriched images were often hard to identify ($\sim$10% of all false negatives). A P-junction looks like two adjacent cells with a bright region in the fluorescent image. This pattern is not unlike countless other locations in the images that capture two adjacent cells, or budding cells. Our hope was that Candescence would learn to associate the presence of P-junctions with the larger bounding box of the pseudohypha itself along with the "nearby" P-start. There is some indication that Candescence has learned this calculus, although there are many P-junctions per instance of pseudohypha and they can be quite distal to the P-start. Last, some cells near the edges of the image were overlooked, such as the example in Fig. 3B, panel iii. Overall, many of the false negatives occur in dense, crowded images similar to Fig. 3B, panels ii and iii.

**Candescence exhibits high classification accuracy.** Here, we restricted attention to objects which were correctly located in the images ($n = 2,104$) and investigated the classification performance of Candescence. In total, there are 382 errors (accuracy of 81.8%) (Table 2). Figure 3C depicts the confusion matrix across the 15 classes in the

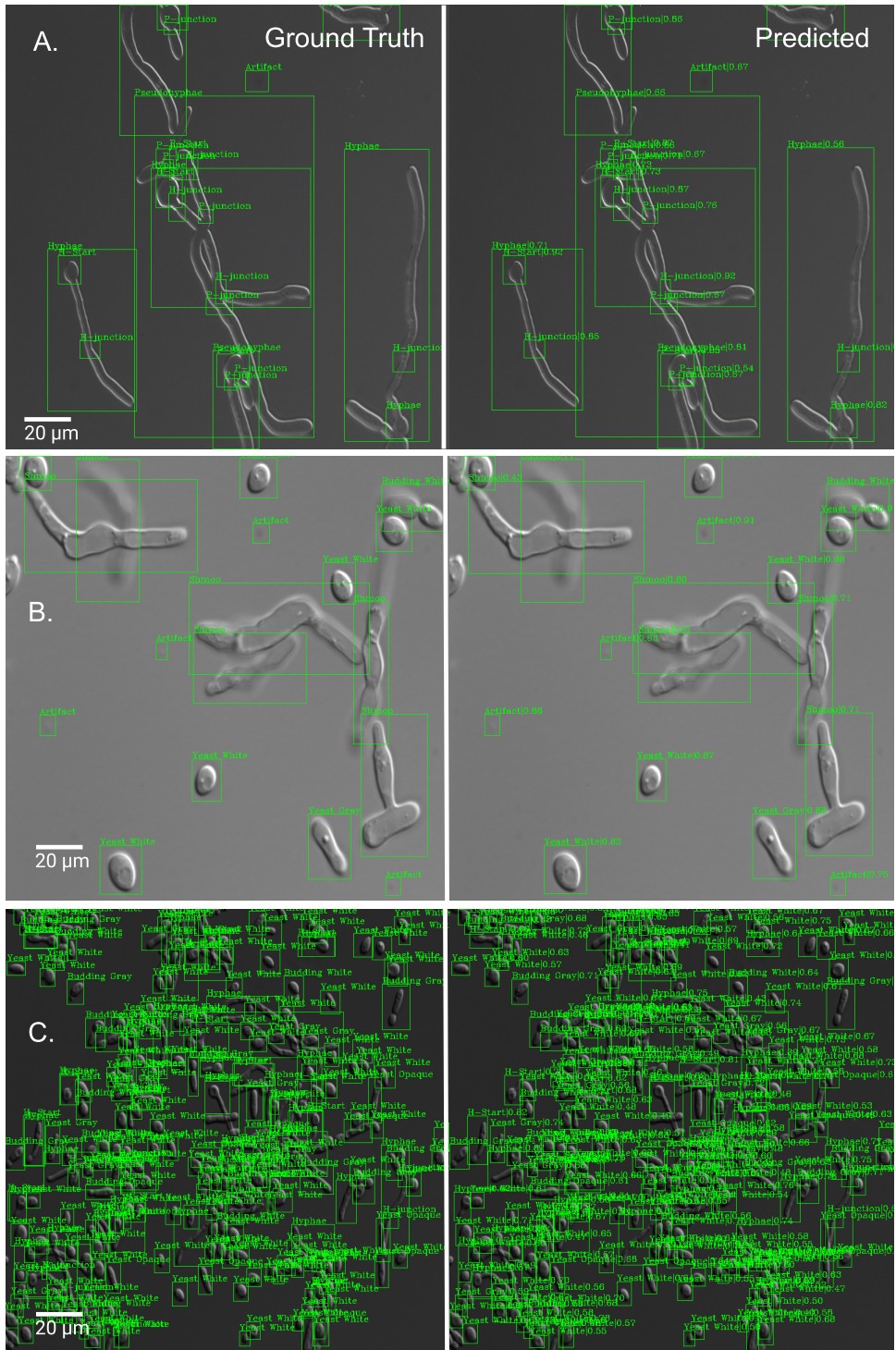

**FIG 2** Representative images of the ground truth in the Varasana validation set (left), each of which consists of a bounding box around the object and a class label, and the predictions made by the FCOS Candescence (right), which also have a score between 0 and 1 from the softmax layer of the FCOS representing the strength of belief in the classification. (A) We found this image difficult to label, as the objects have both pseudohyphal and hyphal properties. Nevertheless, Candescence recapitulated our classifications. Note that if an object is identified as hypha, the junctions and start are also labeled as H; this is also true for a pseudohypha. This suggests that Candescence was learning to classify not only on the image but also as a function of the predicted labels of the same object. (B) Image containing a diverse collection of classes. Note that we labeled gray-like only by size and "texture" (smaller but oblong like opaque and more gaunt than white cells). Overall, these categories witnessed the highest number of classification errors. (C) Candescence predicted well even for highly dense and diverse images such as this one.

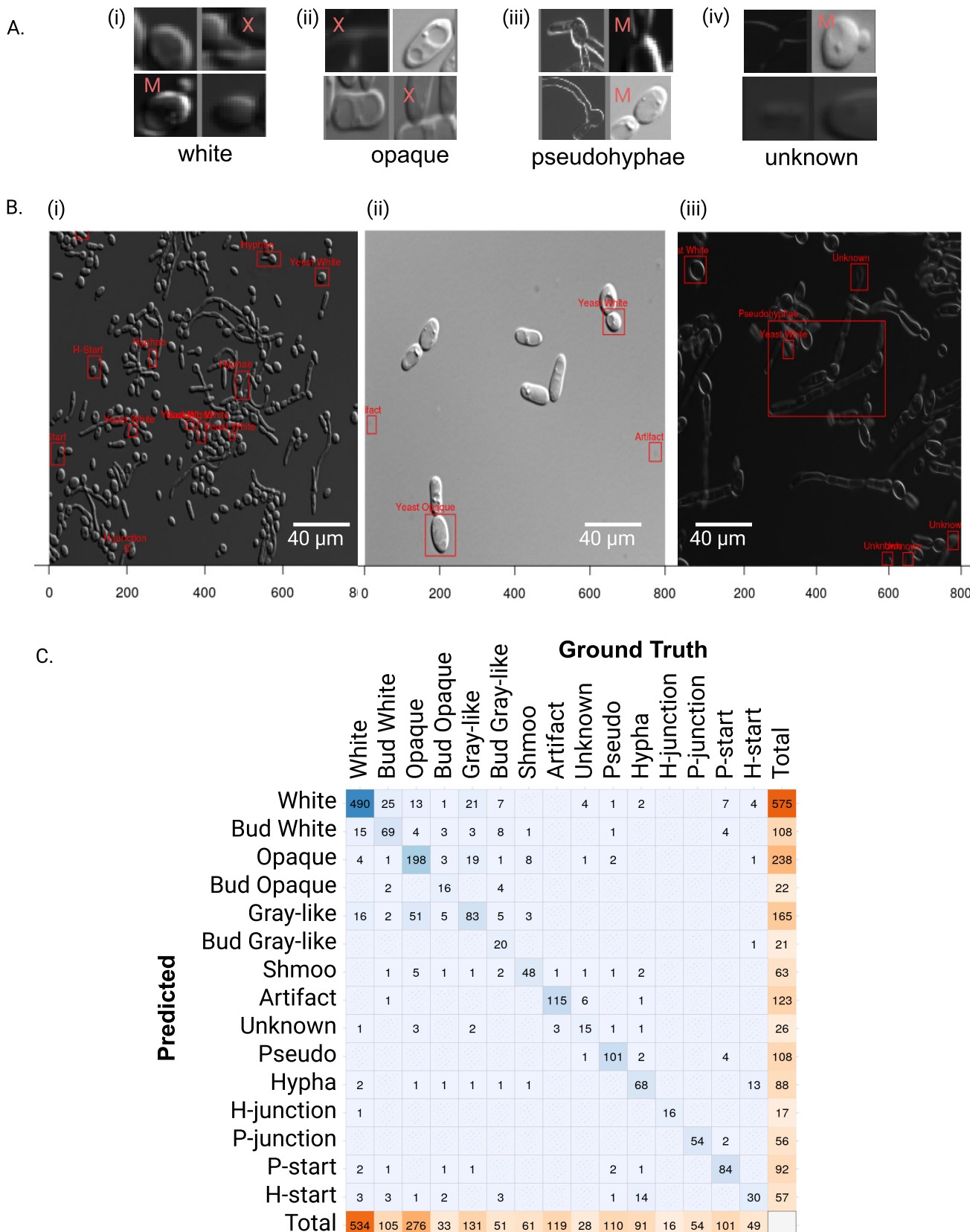

**FIG 3** (A) Representative images chosen from the complete collection in Fig. S3 of all false-positive object identifications from Candescence. This set of hallucinations is subdivided by the class label returned by Candescence. "X" indicates that we truly are not able to see an object (a true false positive/hallucination),

**TABLE 3** Blind spots ($\tau = 0.25$) per class for the best-performing classifier

| Class | Number of blind spots |
| --- | --- |
| Total | 301 |
| Artifact | 77 |
| Yeast white | 74 |
| P-junction | 29 |
| Shmoo | 19 |
| Opaque | 18 |
| Gray-like | 17 |
| Unknown | 16 |
| Pseudohypha | 13 |
| Budding white | 9 |
| Hypha | 7 |
| H-junction | 6 |
| Budding opaque | 4 |
| H-start | 4 |
| Budding gray | 2 |

validation data set, with diagonal entries corresponding to correct classifications. Comparing our labels to predictions, we observe pronounced confusion between opaque and gray-like with some 51 opaque cells classified as gray-like and 19 gray-like as opaque. Significant but slightly reduced confusion exists between white and gray-like. In fact, misclassifications between white, gray-like, and opaque account for almost one-third of all classification errors. Although there is considerable confusion between these three classes, we note that the statistical performance is still far better than random, suggesting that our visual inspection and labeling were at least partially consistent. A smaller degree of confusion exists with hypha and H-start. The softmax appears to diffuse probabilities across H-start, white, and budding white for some bounding boxes. This is perhaps another manifestation of the difficulties we observed with false negatives/blind spots depicted in Fig. 3B.

**Candescence retains its capacity to classify genetically perturbed *C. albicans* in the test set.** Our test set of 904 images serves as an independent measure of performance. Some images of SC5314 or SN148a strains without genetic modifications had been left out of both the training and validation sets, but the strains were grown under the same conditions needed to induce different morphologies. We compared Candescence predictions to manual curation for images referenced in Table 1, but differences in performance between the validation and test set were insignificant (comparison of proportions [$\chi^2$ test]; see "Measures of system performance" in Materials and Methods).

The test also contains *C. albicans* colonies grown with a variety of conditions, preparation protocols, and genetic perturbations (Table 1; Table S1). The four perturbed genes all have well-established and central roles in filamentation processes and the regulation of morphology. Modifications of these genes generated cells that can differ visually from wild-type morphologies, creating an interesting challenge to the ability of Candescence to identify and classify cells.

Our panel includes a transformed SC5314 strain with a CRISPR/Cas9-based homozygous deletion of *UME6* (unscheduled meiotic gene expression), a Zn(II)$_2$Cys$_6$ (zinc cluster) transcription factor that controls transition to true hyphae by maintaining expression of

**FIG 3** Legend (Continued)

and "M" indicates that Candescence was indeed correct to predict a bounding box at that location (missed by the human annotators) but its subsequent classification disagrees with our criteria. Subimages lacking an annotation indicate that Candescence correctly identified and classified the object, which represents a human error. (B) Three partial images with false negatives (blind spots) labeled with red boxes. Unannotated cells in panels i to iii were correctly handled by Candescence, and their bounding boxes have been removed for clarity. (i) Example of a recurrent problem where the probability from the softmax is distributed over two or more labels (e.g., white and budding white), presumably because it has difficulty guessing whether they are only touching versus still attached. The shared probability causes both to fall below our threshold $\tau$. (ii) Although Candescence performed well, blind spots arose presumably due to the dense packing of cells. (iii) Our strategy during labeling was to leave cells that were partially outside the field of view unlabeled. Such edge effects cause some problems and are often predicted as unknown. Last, bounding boxes for large filamentous *C. albicans* cells are sometimes missed, especially if other cells are within the vicinity. (C) Confusion matrix for Candescence. Columns correspond to ground truth classifications, and rows correspond to Candescence predictions.

filament-specific genes in response to inducing conditions. Although cells lacking *UME6* are able to form germ tubes, hyphal extension is limited (60). Our panel also includes biofilm regulatory 1 (*BRG1*), encoding a transcription factor that recruits the histone deacetylase Hda1 to hypha-specific promoters and removes Nrg1 inhibition to promote filamentation. Filamentation is decreased in *brg1Δ/brg1Δ* cells (61). We did not detect a statistically significant difference in performance of Candescence for both *UME6*-null and *BRG1*-null cells under conditions inducing both the white morphology (Fig. S5A) and the pseudohyphal morphology (Fig. S5B).

*RHA1* (regulator of hyphal activity 1) encodes a zinc cluster transcription factor that serves as a regulator of the Nrg1/Brg1 switch. Hyperactivation of Rha1 can trigger filamentous growth in the absence of external signals. In the presence of serum, it can bypass the need for Brg1 (62). Loss of Rha1 function leads to reduced ability to generate hyphal growth in the presence of external signatures. Loss of both Rha1 and Ume6 ablates filamentation completely. Although Rha1-null cells generate yeast white cells with a standard appearance when grown at 30°C in YPD, they produce smallish pseudohyphae with reduced branching when grown at 30°C with serum added to the YPD. Candescence does successfully label pseudohyphal substructures, including P-start and P-junction, but there is some increased confusion with hyphae, as exemplified in Fig. S5C. A minor depreciation in accuracy was observed, and this was statistically significant ($P < 0.05$; type 9) (Table 1).

We cultured *RHA1* gain-of-function (GOF) mutants constructed using the zinc cluster hyperactivation technique of Schillig and Morschhäuser (63) and observed that cells grown at 30°C in YPD formed pseudohyphae that tended to look like the wild type (Fig. S5D). Some of these images were used in the training and validation data sets, and performance remained the same on the omitted test images. However, the performance of Candescence was statistically worse in *RHA1* GOF *UME6*-null cells. Although these cells present with a small pseudohyphal morphology, the images tend to contain multiple tightly packed clusters, which we hypothesize contributes to an increase in the number of blind spots (Fig. S5E).

*RHA1* GOF *BRG1*-null cells are morphologically distinct from wild-type hyphae and pseudohyphae, comprising elongated chains without branching and with less pinching at junctions (Fig. S5F). Here, Candescence most often labeled these cells as hyphae. Interestingly, if an object was labeled as hypha, the junctions were labeled as H-junctions and not P-junctions, although often the characteristic pinching of pseudohyphae is clearly present.

Candescence again experienced a loss in performance with *RHA1* GOF *BCR1*-null cells grown at 30°C in YPD only. Bcr1 is a C2H2 zinc finger transcription factor which regulates **a**/$\alpha$ biofilm formation and cell surface-associated genes. The homozygously null variant exhibited decreased adhesion, biofilm formation, and cell size. There are conflicting data regarding whether it promotes or inhibits filamentous growth; however, cells which do transit to a filamentous morphology appear abnormal (64, 65). In the Varasana image set, this mutant strain generates cells that appear to branch similarly to pseudohyphae, but the individual cells are yeast white or budding yeast in appearance. They tend to form thick clusters in the image (Fig. S6A). The performance decrease is highly significant ($P < 0.001$), with many cells in dense clumps labeled as white or budding white. Candescence did, however, identify the location of the vast majority of cells and often correctly labeled isolated objects as pseudohyphae. When *RHA1* GOF *BCR1*-null cells were grown at 37°C with serum added to the medium (types 13, 17, and 20) (Table 1), we observed large pseudohyphae, and Candescence classified correctly at the same rate as the validation set (Fig. S6B).

**Capturing the canonical forms of *C. albicans* morphologies: generative adversarial networks.** We observed considerable cell-to-cell morphological variability across the images. Heterogeneity arises due to technical variations (e.g., light intensity and focus), natural biological programs (e.g., cell cycle affecting size/shape), and transitions between morphologies (e.g., growth of a hyphal cell from a germ tube). These sources of heterogeneity complicate both the manual labeling procedure during construction

of the learning set and the downstream ability of an FCOS to correctly assign morphology. We studied this variability first using an unbiased, unsupervised variational autoencoder (VAE) (66) approach (Fig. S8). We reasoned that if a collection of images is sufficiently large, it will likely capture snapshots of cells in all technical (e.g., different light intensities), developmental (e.g., across each step of the cell cycle), and morphological states (e.g., along the transition from germ tube to hypha). Using Varasana as a first approximation to this large collection, our goal here was to build pseudo-time models that capture the progression of these technical and biological variables. Our approach is based on generative adversarial networks (GANs). Intuitively, this deep learning technique rerepresents the images in a latent space in a manner that captures these progressions (or "trajectories"). Then, computational techniques can search for trajectories through the latent space that correspond to a specific effect of interest (e.g., a specific transformation between two morphologies). This allows us to derive continuous "movie-like" models of cells morphing along this trajectory.

GANs are trained using a game-theoretic adversarial approach that pits two deep networks—the generator and the discriminator—against each other (67). The purpose of the generator model is to create "fake" examples of *C. albicans* cells with different morphologies. These fake images are created to deceive its adversary, the discriminator model. The generator is allowed to feed both fake and real images to the discriminator model, whose goal is to differentiate between the two. The generator is then told which images the discriminator got right or wrong; this information is used to update the generative model (which corresponds to updating parameters of the neural network). This creates an "arms race" between the deep networks. After a sufficient number of epochs, the generator will ideally produce fake images of *C. albicans* morphologies which cannot be distinguished from true images by the discriminator. Our GAN is based on the computationally accessible method from Liu and colleagues (68).

After training, we exploited the generator to explore trajectories in our latent space. We started with two images representing the endpoints of a process of interest. For example, the left endpoint might be a real image of a yeast white cell, while the right endpoint might correspond to an opaque cell. We then used a process called inversion (69) to find a representation of these two images in the latent space of the generator (see "Generative adversarial networks" in Materials and Methods). Figure 4A depicts examples of target images and their nearest neighbor in the latent space. Next, the system finds a linear path between these two points in the latent space, so that the nearest neighbor of the final "fake" image at s_8 is the real image at the right endpoint. Last, the intermediate "fake" images are reconstructed from the latent space to provide a visualization of the trajectory using the generator function.

Figure 4B (panel i) depicts the results of applying this procedure to find the yeast white-to-opaque morphological switch. Interestingly, the system seems to arrive at a decision point at $s_3$. At this point, the trajectory could bifurcate toward budding white, although in this case it continues toward opaque. As a comparison, when we asked for a trajectory from yeast white to budding white (Fig. 4B, panel ii), the trajectory stays true to the yeast form and does not appear to move toward the opaque morphology. The hypothesized bud is perhaps somewhat disproportionately large compared to its mother and larger than the bud of the real image. Figure 4B (panel iii) presents a trajectory that starts with a budding opaque cell. Both cells appear to bud a second time from $s_4$ through $s_6$. The progeny of the daughter cell (left) disappears from the trajectory. Although imperfect, the system appears to have learned a reasonable model of pseudohyphal development from relatively few (~100) images. Continuous movies for each of these trajectories are available from our on-line resources.

**Detecting deviations from standard *C. albicans* morphologies: anomaly detection with GANs.** Microscopy is routinely used to judge whether a specific genetic or environmental perturbation has led to an observable phenotype. This could manifest as a visual change in the composition of cells in an image (e.g., an increase in opaque cells versus control), a difference in the spatial distribution of cells in the image (e.g., clumping of cells), or a change in appearance of the cell (e.g., shortened hyphae or

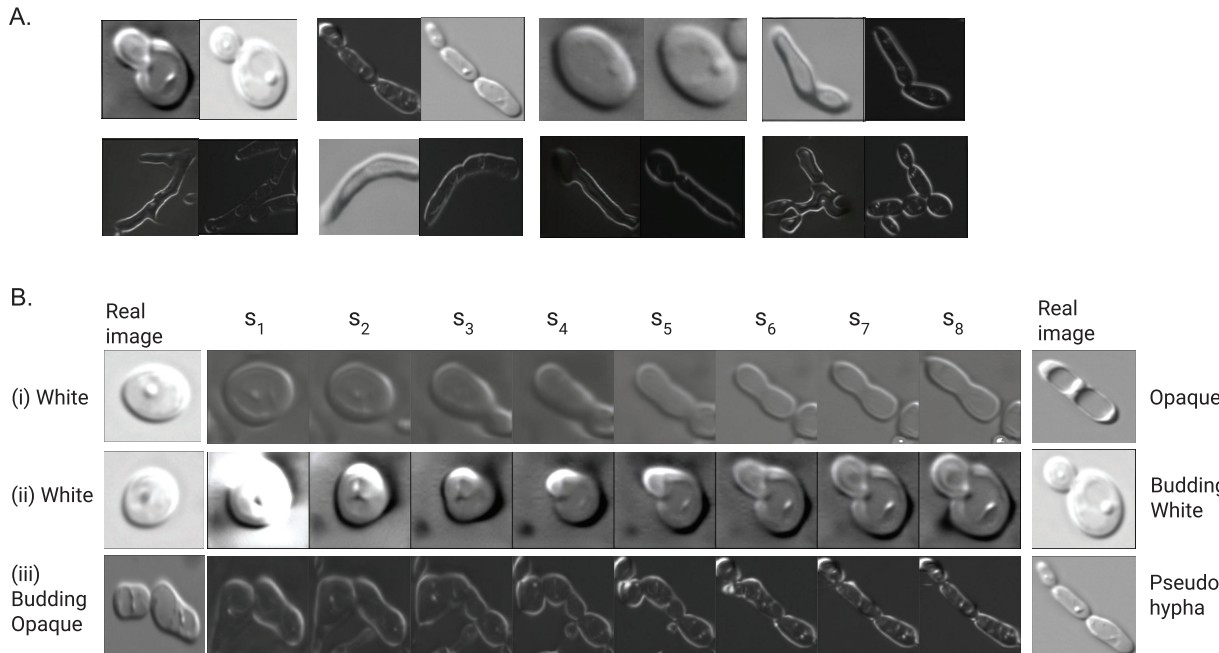

**FIG 4** (A) For each pair of images, the left image is a synthesized image produced by our model, and the right image is the image found from the real training data that is its nearest neighbor under the LPIPS metric. (B) Three separate trajectories. Here, the left- and rightmost images correspond to real images from the training data. The sequence of images at points $s_1$ to $s_7$ are produced by a linear interpolation between the real images in the generator's latent space in the GAN.

large vacuoles). Next, we built upon our GAN-based morphology models to address the third issue, namely, a system capable of deciding whether the cells in an image deviate significantly from the space of wild-type *C. albicans* morphologies. Our hope is that the deep learner is more sensitive than "eyeballing" microscopy images when manually attempting to investigate if an experimental strain is abnormal.

The algorithm starts by mapping a target bounding box (representing a single cell, hypha, or pseudohypha) into the latent space of the GAN's generator (see "Anomaly detection" in Materials and Methods). The nearest neighbor in the latent space is found; this corresponds to the object from the training and validation data set that is visually most similar to the target. The distance between the target cell and nearest neighbor is computed using a specialized similarity metric developed from the learned perceptual image patch similarity (70) measure. The intuition is that distances between a target with a normal morphology will have a nearest neighbor that is closer in the latent space than a target with a very abnormal morphology.

Figure 5 presents an example of this test with a *RHA1* GOF *BCR1*-null colony. Candescence was first used to identify the objects and their morphology in the image, and the anomaly score was computed for each such object in the latent space built from the Varasana learning set. When we compared the distribution of anomaly scores between all images from this mutant colony (type 13) and compared them against a collection of "wild-type" pseudohyphal cells (type 65), we observed a statistical enrichment of outliers with abnormal morphology (Kolmogorov-Smirnoff test, $P < 0.01$). We generally did not observe differences at the low end of anomaly scores, as almost all images from genetically perturbed colonies still contained many examples of cells with normal morphology.

## DISCUSSION

Candescence includes a multiobject detection algorithm capable of accurately classifying nine *C. albicans* morphologies. It is based on a fully convolutional one-stage (FCOS) architecture which both locates objects and classifies them with high accuracy.

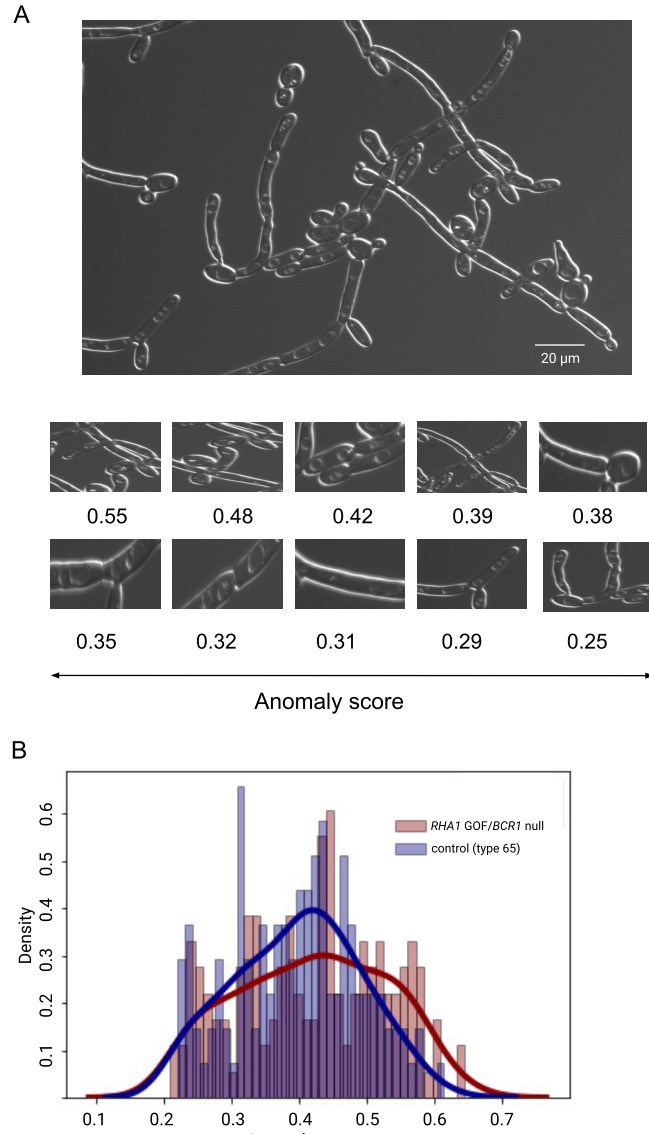

**FIG 5** (A) Example of anomaly detection using a *RHA1* GOF *BCR1*-null colony (UID 230 of type 13). The panels at the bottom are enlargements a series of bounding boxes predicted by Candescence with their associated anomaly score. (B) Histogram comparing the distribution of anomaly scores between all images of type 13 versus type 30, a collection of normal-appearing pseudohyphal and hyphal cells. There is a small but statistically significant enrichment of cells from the *RHA1* GOF *BCR1*-null colony with elevated anomaly score differences (Kolmogorov-Smirnoff test, $P < 0.01$).

The system was trained with the Varasana learning set consisting of ~1,200 total images. Using transfer learning, the starting point for training was ResNet-101, a network capable of locating and classifying common household items and pets. It is possible that the image building blocks (textures, edges, and colors) encoded in ResNet-101 are not optimal for (DIC) microscopy. As our collection of *C. albicans* images grows, it may be feasible to build a microscopy-specific analog of ResNet-101 and improve performance.

Since a flat-structured learning set led to suboptimal performance, we developed a six-grade curriculum learning set and ordered examples by increasing difficulty. To the best of our knowledge, we are the first to use a cumulative approach with a curriculum strategy where images appear at one grade and reappear in all subsequent grades. We hypothesize that this approach removes the need for layered freezing strategies. This

is advantageous, since optimization of hyperparameters is computationally expensive and the space of possible freezing strategies grows exponentially in the number of grades and network layers.

Candescence exhibited very good performance. Given that an object was correctly regressed in the image, it was assigned the correct class label with ~82% probability. Misclassifications tended to occur between morphologies that overlap, as highlighted by the VAE plots of Fig. S8. Confusion tended to exist between vegetative morphologies and their budding/mating forms (e.g., from white yeast to budding white). The difficulty here for Candescence was to judge cases where the daughter cell was large but still attached versus detached but still physically adjacent to its mother. We hypothesize that a larger set of training examples will remove this confusion. Inclusion of time-lapse images of specific biological processes would also improve performance. Moreover, integration of the matched fluorescent images with the current grayscale DIC image would perhaps provide the learning procedure with the necessary information to judge whether separation has occurred. It would be straightforward to encode the fluorescent image as an added dimension beyond the grayscale 800-by-800 input currently used. We did not pursue this avenue in the first version due to complications in the training procedure, since some fluorescent images were not available.

The construction of any learning set requires consistent labeling rules. In our setting, there were visible differences in sizable subpopulations that stretch from yeast white to opaque. This includes a large number of small rectangular "gaunt" cells we labeled as gray-like. Candescence was at times confused between the yeast white, gray-like, and opaque morphologies, an expected observation given that there were many images for which we were unable to form consensus as a group of humans. Absolute assessment of true gray versus our gray-like was out of reach for this study, as we lack the necessary molecular markers as per the original findings of Tao and colleagues (15). Although false classifications are enriched between white, opaque, and gray-like, the success rate is still very good, suggesting that this dichotomy does exist in the images. If these cells had instead been randomly assigned the three class labels without regard to physical appearance, it is highly unlikely that a classifier could learn to distinguish them with statistically significant performance. Although the relationship to the gray cells reported by Tian et al. (56) remains unsettled, it does suggest that there is an interesting substructure across the vegetative forms that is perhaps not captured by our current dichotomy; other cryptic morphologies could exist. These physically distinct subcolonies could represent simply canonical mating-competent forms at specific moments of their development or phenotype diversity that arises in response to an environmental or communal cue (71). Building upon this primeval version, Candescence may eventually allow the interrogation of community structure and interaction.

Images of stained or fluorescent reporter molecules would help resolve issues of cryptic morphologies and would extend the capacity of the system to classify using subcellular features. This technique has been used successfully to explore changes to *S. cerevisiae* in the presence of genetic perturbations (53), although baker's yeast does not have as large a range of morphologies as *C. albicans*.

Our system initially appeared to have significant challenges during the object detection step. However, careful analysis of the false-positive detections suggests that many such events are in fact not hallucinations (false positives) but "false false positives." That is, they correspond to true events in the image files that we missed during the manual annotation procedure. A large portion of these events correspond to either subtle technical artifacts or small cells in crowded images.

Candescence appears to have some blind spots, missing cells that are annotated in the ground truth data set. The trade-off between false negatives and positives is controlled by underlying IoU and threshold $\tau$ parameters. We observed that accurate bounding boxes were indeed regressed for the majority of these false-negative objects, but such objects required an abatement of either the IoU or $\tau$ parameter

before they were reported as positive predictions. Our forensic analysis of the neural network suggests that this is due to confusion between, for example, yeast white and yeast budding white; the probability is amortized over two or more classes and therefore drops below $\tau$.

Furthermore, with respect to false-negative objects, we hypothesize that the differences in size and shape between the morphologies (e.g., yeast white versus pseudohypha) induce different distributions of IoU and $\tau$ scores. Therefore, in images containing diverse cell types, the single universal $\tau$ is essentially too conservative for some morphologies but too liberal for others. It is an interesting future challenge to modify FCOS-based classifiers to adjust for this heterogeneity in a statistically sound manner.

We explored in the independent test set a range of genetically altered *C. albicans* populations involving the genes *RHA1*, *UME6*, *BCR1*, and *BGR1*, which have established roles controlling filamentation. The observed changes in classification accuracy are consistent with the fact that these morphologies represent shifts away from wild-type forms. The Candescence response to these perturbations was intuitive, and it largely retained its performance.

Our work can be viewed as a revisitation and extension of the quantitative models from 1989 introduced by Merson-Davies et al. (54, 55) using modern, powerful deep encoders. Our intention is that the GAN models can be used to automatically detect new morphologies that are perhaps subtly different from our current dichotomy. Using tools for detecting anomalies via the generator's latent space, we show how cells displaying noncanonical morphological forms can be flagged and quantified. This should find practical value in microscopy-based studies: the tool will provide the community with a central resource that houses not only all *C. albicans* images but also unbiased models developed from those images that extended to noncanonical morphologies. Future studies will benefit from the ability to compare their images across this synthesized sum of knowledge. The technique should be straightforward to transfer to other fungi.

We have shown that the deep learning-based approaches are able to recapitulate the classification rules that are encoded by our choice of labeling strategy. It is unlikely that our labeling strategy is correct, and other labelers with more or different expertise might have chosen a different way to partition the learning set and assign labels to individual examples. Varasana version 1.0 labels are certainly imperfect and open to debate, but the classifier does function far beyond random guesses, suggesting that our scheme has some value.

There is evidence that Candescence is able to "overcome" errors and inconsistencies between labelers. This is a well-documented problem in image recognition research, including computational pathology (72). We argue that the computational techniques introduced to computational pathology are largely applicable to fungal systems. For example, machine learning-based analysis of medical images has been shown to be more sensitive than trained pathologists identifying small events and complex patterns beyond perhaps human capacities (73). As Candescence evolves to include stainings, fluorescent markers, and a greater spectrum of microscopy imaging techniques, it may reveal cryptic cell or subcellular structure or community organization. Computational approaches in imaging provide a means to combine different modes of data and to provide downstream analyses that integrate this information in a statistically sound manner (74–76). For *C. albicans*, this might entail the integration of information concerning strain, growth conditions, and genetic challenges together with images to better understand the composition and dynamics of colonies. Imaging standards analogous to Digital Imaging and Communication (DICOM) (77) for microbial systems, including imaging of host tissue, would enable better data sharing across the fungal community and allow "hive analysis" (78).

Last and perhaps most importantly, a surprising degree of disagreement has been observed (and quantified via Cohen's $\kappa$ score) between expert pathologists when challenged with the same images (79). Our limited experience with *C. albicans* morphology

suggests that there may be similar disagreements among experts in this field. Such differences may hint at important alternative classification schemes. These differences may be important as Candescence is extended into clinical samples where the presence of *C. albicans* and its morphology are considered concomitantly within their host tissue. Our effort here represents an opportunity for the community to kernelize their knowledge of the dynamics of morphologies in a quantitative objective manner.

## MATERIALS AND METHODS

**A fully convolutional one-stage object detector for morphology classification.** There are fundamentally two computational problems underlying multiclass, multiobject detection. The first problem is to detect the locations in an image where objects exist; this is a regression problem where we need to determine the four coordinates corresponding to the corners of the bounding box. In our case, the number of objects per image ranges up to ~100 (Fig. S1), and objects may overlap in these images, especially with respect to the filamentous morphology. Most object detectors rely on predefined anchor boxes. An anchor box is a rectangle that bounds an object in an image. Approaches that use anchor boxes make educated guesses as to where objects might exist in the image in addition to guesses regarding the size, aspect ratio, and number of such boxes. The fact that there are exponentially many potential anchor boxes in any image makes this a computationally demanding exercise. Moreover, there are many parameters (size, aspect ratio, and number of boxes) that require optimization and redesign on new data sets. The second problem is then to correctly label each object by its class (13 classes representing 9 morphologies, an unknown class, and an artifacts class).

Here, we opted to use a fully convolutional one-stage object detector (FCOS) for classifying *C. albicans* morphology (56). FCOSs represent an anchor box-free reformulation of object detection. This is achieved by predicting, for each point in each feature map, the offset position to the top left and bottom right coordinates of a bounding box. Five feature maps are produced, and each such map is limited to predicting bounding boxes of a predetermined size. For example, the first feature map predicts bounding boxes with maximum area of 30 by 30 pixels, while the final feature map is used to predict bounding boxes with maximum area of 128 by 128 pixels. A standard convolutional neural network is used with a softmax head for the classification component. We developed Candescence on top of the open-source implementation of FCOS provided by the machine vision platform MMDetection (80) (Fig. 1F).

**Strains and media.** Images of yeast white, opaque, and shmoo morphologies were acquired from *C. albicans* SN148a cells grown on YPD agar (1% yeast extract, 2% Bacto peptone, 2% D-glucose, 2% agar, and 50 $\mu$g/mL uridine). Opaque switching was induced by growing cells on SC glucosamine medium (0.67% yeast nitrogen base lacking amino acids, 0.15% complete amino acid mixture, 2% agar, 1.25% *N*-acetylglucosamine [GlcNAc], 100 $\mu$g/mL uridine). We used 5 $\mu$g/mL phloxine B to stain opaque colonies. The shmoo morphology was induced by treating opaque cells with 10 $\mu$g/mL $\alpha$-pheromone for 24 h in room temperature shaking at 220 rpm.

To induce filamentation in wild-type SC5314, two colonies of cells was grown separately in 5 mL of glucose-phosphate-proline (GPP) medium (81) (2.5 mM $KH_2PO_4$ [pH 6.5], 10.2 mM L-proline, 2.6 mM *N*-acetyl-D-glucosamine, 3 mM $MgSO_4 \cdot 7H_2O$, 20% glucose) for 12 to 16 h in a 30 and 37°C shaker incubator. The next day, 1 mL of cells from each colony was washed twice with 1 mL $1\times$ phosphate-buffered saline (PBS). Different genetic variants of the SC5314 strain were also used to generate colonies enriched for filamentous morphologies. After growth, cells were washed with $1\times$ PBS and diluted to different $OD_{600}$ values in fresh liquid spider medium, YPD, or combinations of YPD and fetal bovine serum with or without centrifugation as per Table S1, which provides a complete list of strains, conditions, and protocols.

**Microscopy.** *C. albicans* colonies were mounted on slides and stained with a concentration of 2 $\mu$g/mL calcofluor white for 20 min before imaging. Images of *C. albicans* were captured using a Leica DM6000 upright microscope equipped with $100\times$ (numerical aperture [NA], 1.3), $60\times$ (NA, 1.4), and $40\times$ (NA, 0.75) lenses and a Hamamatsu Orca ER camera. For DIC images, samples were captured using DIC optics and the built-in transmitted illuminator of the microscope. For cells labeled with fluorescent probes, samples were illuminated with a 100-W mercury bulb (Osram) and passed through filter cubes optimized for illumination of calcofluor white-labeled samples (excitation wavelength, 377/50 nm; emission wavelength, 447/60). We used the fluorescent images during the manual labeling procedure to help decide on the best morphological assignment. This was particularly relevant to distinguish between white/opaque/gray-like and their budding variants, and also between junction types for the filamentous morphologies, as bud scars are clearly visible. However, the fluorescent images were not submitted to the FCOS (or other deep learning tool) during training. They are available as part of the Varasana learning set.

**Image annotation and development of the learning set.** In general, all computations were done using Python version 3.7 and R version 3.6.3. Our learning set was prepared using Labelbox (https://labelbox.com), software that facilitates the distributed annotation of image files. As a group, we labeled images following the guidelines from Sudbery et al. (20), Whiteway and Bachewich (11), Noble et al. (22), and Tao et al. (15). Labelbox assigns images in a manner that guarantees that the same image is scored by multiple labelers. One labeler (M.T.H.) modified assignments after the first round of labeling to ensure as much consistency as possible across the labelers. A second round of quality control was performed by M.T.H. after the grid search was completed and our best classifier identified. Here, all false positives

and negatives were examined, and a decision was made as to whether the instance was a labeling mistake or a mistake made by the classifier.

**Measures of system performance.** In the expressions below, TP, TN, FP, and FN denote true positives, true negatives, false positives, and false negatives, respectively. Recall (also called sensitivity) measures the rate of false negatives: TP/(TP + FN). Precision measures the rate of false positives: TP/(TP + FP). The $F_1$ measure is convenient, as it combines recall and precision into a single summary statistic: {TP/[TP + 1/2(FP + FN)]} + 2/(recall$^{-1}$ + precision$^{-1}$).

In multiobject/multiclass problems, there are two fundamental parameters. The first parameter is related to the object detection component of the FCOS and is termed the IoU value, calculated as area of intersection/area of union. Throughout this work, we used a threshold of 0.5 for the IoU. The IoU controls how closely the predicted bounding box must overlap the ground truth bounding box to be considered a positive. More stringent IoUs tend to decrease the recall of the system significantly, with recall dropping due to a rapid increase in the number of false negatives (82).

The second parameter is related to the classification component of the FCOS. Here, $\tau$ represents the minimum value from the softmax of the classification head of the FCOS that must be exceeded if an object is to be assigned a class. More precisely, the object is assigned a class if and only if (i) the class has the highest score and (ii) the score exceeds $\tau$.

The dual nature of object detection/object classification problems requires refinement of these fundamental concepts. We say that an object in an image is a TP if and only if the bounding box is predicted correctly (i.e., the IoU is >0.5) and the object is then classified correctly: $\kappa = \text{argmax}_{\text{class } c} \; \text{score}(\sigma, c)$, where $\kappa$ is the true class of the target $\sigma$ and score$(\sigma,c) > \tau$.

The concept of a true negative in this setting is tricky, since any pixel which does not belong to a bounding box in an image is in essence a TN. An object is an FP if and only if either (i) the predicted bounding box does not sufficiently overlap a ground truth bounding box or (ii) the predicted bounding box does sufficiently overlap a ground truth bounding box but the classification is incorrect. Type 1 is termed a hallucination; type 2 is a misclassification. An object is an FN if and only if we fail to predict a bounding box with sufficient overlap a ground truth bounding box. We term these FNs blind spots.

The mean average precision (mAP) is the standard and preferred approach for measuring the performance of multiobject/multiclass problems (59). The mAP computes the average of the average precision over $\tau$. Here, the average precision corresponds to the area under the precision recall curve induce by a specific threshold for the IoU and varying $\tau$. The maximum value for the mAP is 1.

Within an FCOS, total loss is computed as the sum of three individual losses: centerness, bounding box, and classification loss (Fig. S2). The concept of centerness loss is specific to the FCOS and represents a mechanism to avoid the identification of multiple, spurious bounding boxes for a single object. It converges to a value of 0.57 (56, 83). Bounding box loss measures disagreement between the ground truth location of bounding boxes with the regression produced by the deep learner. Finally, classification loss measures how well correctly identified objects are assigned classes.

We searched the hyperparameter space defined by the cross-product of different settings for the learning rate, momentum, decay, number of epochs per grade, IoU, threshold $\tau$, and various freezing rates (Fig. 1H; boldface font denotes the choice of parameters for the final FCOS). Throughout these experiments, the 800-by-800 input images were subjected to augmentation (random multiscale flipping). Other parameters included 1,000 warmup iterations, a warmup ratio of 1/3, and an SGD optimizer with grad clipping. The mAP and three loss functions described above were used to initially judge the quality of the classifier.

When investigating differences in performance of a classifier between the test and validation sets, we manually curated a subset of test images for each entry in Table 1 in a manner to ensure that at least 100 cells were labeled (with the exception of the first two entries of Table 1, where there were too few cells). Then we built a contingency table where rows correspond to correct and incorrect predictions and columns correspond to the test and validation data set. A $\chi^2$ test was used with the null hypothesis of no difference between the overall number of correct predictions in the test and validation sets.

**Generative adversarial networks.** We followed the approach of Liu et al. (68) to build unconditional GAN images. This approach requires computational resources that are accessible to most labs and requires few training samples, a feature important for our setting here. The model is particularly amenable to the disentanglement procedures described below. Figure S7A and B depict the structure of the generator and discriminator, respectively. We extended the Pytorch-based code available from the authors (84). The input used for training corresponds to the ground truth bounding boxes of the training and validation data sets, reshaped as a 128-by-128 image. Here, we used a shift predictor and deformator learning rate of 0.0001, with 2,000 steps in batches with a size of 4.

We followed the framework of Creswell and Bharath (69) to build trajectories between two (real) target images. Here, however, we opted to use the learned perceptual image patch similarity (LPIPS) metric as the loss function (70). The core idea is to find both "real" target images in the generator's latent space. This requires a so-called inversion, which we perform using the algorithm INFER of Creswell and Bharath (69). Then, we used linear interpolation between these two points in the latent space and reconstructed visual representations at user-defined points along this path.

**Anomaly detection.** Anomaly detection proceeds using the framework from Schlegl et al. (85) but replacing their residual and discriminator loss with the LPIPS metric for estimating the similarity between two images (70). Our approach proceeds as follows. (i) Each object in an image file is bounded manually, or Candescence is used to automatically regress bounding boxes for the objects in the target image file. (ii) For each target object (that is, for each bounding box or patch) *T*, we found its optimal inversion *z* in the latent space of the generator *G* using the INFER(*T*,*G*) algorithm of Creswell and Bharath

(69) modified to use the LPIPS similarity metric. Here, $G(z)$ is a synthetic image. (iii) Next, we found the nearest real neighbor $d$ of $G(z)$ across all images $d$ in the training set $D$ under the LPIPS metric. (iv) The anomaly score is then defined as follows: $A(T|G,D) = L[G(z),T] + \min_{d \in D} L[G(z),d]$, where $D$ is the set of training images and $L$ is the LPIPS function between two images.

With respect to the second part of item 1, we remark that the performance of Candescence was not severely reduced when given an image with noncanonical morphologies, and in this setting, the classification returned by the FCOS is not used. Therefore, this automated approach should suffice in most scenarios, unless the change in morphology is very large. However, when the change in morphology is obvious, we will not need a sensitive algorithm to detect it.

**Data availability.** The Varasana data set, code, trained models, supplemental movies, and a Jupyter notebook to run Candescence are available from the Open Science Framework at https://osf.io/qdxbp.

## SUPPLEMENTAL MATERIAL

Supplemental material is available online only.
**SUPPLEMENTAL FILE 1**, SVG file, 0.5 MB.
**SUPPLEMENTAL FILE 2**, SVG file, 0.5 MB.
**SUPPLEMENTAL FILE 3**, SVG file, 0.5 MB.
**SUPPLEMENTAL FILE 4**, SVG file, 0.3 MB.
**SUPPLEMENTAL FILE 5**, SVG file, 7.5 MB.
**SUPPLEMENTAL FILE 6**, SVG file, 2.5 MB.
**SUPPLEMENTAL FILE 7**, SVG file, 0.1 MB.
**SUPPLEMENTAL FILE 8**, SVG file, 2.2 MB.
**SUPPLEMENTAL FILE 9**, XLSX file, 0.2 MB.
**SUPPLEMENTAL FILE 10**, PDF file, 0.1 MB.

## ACKNOWLEDGMENTS

This work was supported by Canadian Research Chair Tier I awards to M.W. and M.T.H. and an NSERC Discovery award to M.T.H.

V.B. and M.T.H. developed the deep learning algorithm, performed experiments, and helped prepare the manuscript. A.C.B.P.C., R.P.O., and S.M. constructed the library of microscopy images used for training. E.K. and S.S. developed the Varasana learning set from these images. C.L. provided expertise in microscopy and analysis of the images. V.D., M.W., and M.T.H. prepared the manuscript. M.T.H. designed the study and obtained funding for the project.

We declare no conflict of interest.

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
