## [Reviewer comments · Microbiology Spectrum]

Microbiology Spectrum

A deep learning approach to capture the essence of *Candida albicans* morphologies

Van Bettauer, Anna Costa, Raha Omran, Samira Massahi, Eftyhios Kirbizakis, Shawn Simpson, Vanessa Dumeaux, Christopher Law, Malcolm Whiteway, and Michael Hallett

Corresponding Author(s): Michael Hallett, University of Western Ontario

Review Timeline:

Submission Date:	April 26, 2022
Editorial Decision:	May 17, 2022
Revision Received:	July 6, 2022
Accepted:	July 25, 2022

Editor: Rebecca Shapiro

Reviewer(s): The reviewers have opted to remain anonymous.

Transaction Report:

DOI: <https://doi.org/10.1128/spectrum.01472-22>

May 17, 2022

Dr. Michael T Hallett
University of Western Ontario
Biochemistry
Medical Science Building M359
London, Ontario N6G 2V4
Canada

Re: Spectrum01472-22 (A deep learning approach to capture the essence of *Candida albicans* morphologies)

Dear Dr. Michael T Hallett:

Thank you for submitting your manuscript to Microbiology Spectrum. As you will see, two reviewers were generally enthusiastic about this work, and have suggested some mostly minor modifications that will need to be incorporated to consider it for publication. When submitting the revised version of your paper, please provide (1) point-by-point responses to the issues raised by the reviewers as file type "Response to Reviewers," not in your cover letter, and (2) a PDF file that indicates the changes from the original submission (by highlighting or underlining the changes) as file type "Marked Up Manuscript - For Review Only". Please use this link to submit your revised manuscript - we strongly recommend that you submit your paper within the next 60 days or reach out to me. Detailed instructions on submitting your revised paper are below.

Link Not Available

Sincerely,

Rebecca Shapiro

Journals Department
Reviewer comments:

Reviewer #1 (Comments for the Author):

This is an interesting technical paper that describes a new image analysis tool -Candescence that can discriminate between none nine morphologies generated by the human pathogenic fungus *Candida albicans*. The papers is reasonably clearly written and the utility of the tool is demonstrated in the analysis of *Candida* mutant phenotypes. There are some loosely, and informally written sections that could be sharpened and the paper misses some key foundation information in the literature that could and should be incorporated. The micrographs need space bars and the manuscript in general needs to be carefully checked for

precision and consistency. It is a very long narrative and in my view this shortening would make it sharper and more accessible. But, this is a helpful addition to the tool kit that *Candida* biologists can use in the future to analyse growth, morphogenesis and mutant phenotyping.

General and specific comments:

1. L 37-39 Abstract: " we then develop models using generative adversarial networks and identify subcomponents of the latent space which control technical variables" won't be understood by *Candida* biologists."? In general, if the specialists computational terminology could be explained carefully that would be helpful.
2. L53 If morphology index is a continuum how do you define break points? I see this is addressed later in the manuscript (L455-465). Important - it should be noted that this problem has been addressed previously by Odds et al (see below) who generated a very simple and useful morphology index.
3. L74. Forms related to mating are rare in mucosal or skin surfaces.
4. L78 I would not describe opaque cells as rectangular
5. L96-98 presentation. "Pseudohyphae tend to have more branch points, because mother-daughter attachments are more easily disrupted than in hyphae..". I'm not sure this is the right interpretation. Pseudohyphae tend to be synchronously dividing and that creates more evaginations that are seen in true hyphae where sub-spacial compartments are often cell cycle arrested and do not re-enter the cell cycle for several generations after they were created. (i.e. the number of branches is not due to breakage as suggested here). Hyphal branching frequency in hyphae is regulated by nutrient composition which affects how long a compartment remains arrested in the cell cycle, (see Sudbery et al 2011, Growth of *Candida albicans* hyphae, *Nat Rev Microbiol*; PMID: 19347730; Sudbery, P., Gow, N.A.R. & Berman, J. (2004). The distinct morphogenic states of *Candida albicans*. *Trends in Microbiology* 12, 317-325. Veses, V & Gow N.A.R. (2009). Pseudohypha budding patterns of *Candida albicans*. *Medical Mycology* 47, 268-275.; and Barelle, C.J., Bohula, E.A., Kron, S.J., Wessels, D., Soll, D.R., Schäfer, A., Brown, A.J.P & Gow N.A.R. (2003). Asynchronous cell cycle in true hyphae of *Candida albicans* is related to asymmetric vacuolar inheritance. *Eukaryotic Cell* 2, 398-410.
6. L103. Chlamypospores are very common on the appropriate media - but they are not usually seen in clinical samples.
7. L104. The work by Odds et al who devised a "morphology index" is perhaps the closest and most relevant early work to this publication, but it is not cited. See L A Merson-Davies 1, F C Odds. *J Gen Microbiol* . 1989 Nov;135(11):3143-52. doi: 10.1099/00221287-135-11-3143. A morphology index for characterization of cell shape in *Candida albicans*. PMID: 2693594; DOI: 10.1099/00221287-135-11-3143. and L A Merson-Davies 1, V Hopwood, R Robert, A Marot-Leblond, J M Senet, F C Odds. 1991 Dec;35(6):321-4. *J Med Microbiol* doi: 10.1099/00222615-35-6-321. Reaction of *Candida albicans* cells of different morphology index with monoclonal antibodies specific for the hyphal form. PMID: 1753389; DOI: 10.1099/00222615-35-6-321.
8. L190. Does this mean the total magnification or is this the objective magnification - the total magnification includes the objective mag and any further magnification of the image on the page.
9. Figure 1. Singular and plural of hypha/ hyphae/ pseudohypha/ Pseudohyphae e.g. the "the start of the pseudohyphae". e.g. Figure 5B(iii) Check throughout the manuscript.
10. Have a space between the units and values - e.g. L 196. L214, L215 etc
11. The morphology of any growth form will also depend on the growth medium. It would be useful to explore the effect of nutrient status, temperature etc on the morphology of key cell types.
12. L311. "Candescence almost never hallucinates but has some blind spots." Use of "hallucinates" is amusing, but may lead to confusion. "generates false predictions" is better. L387 - mutants (brg1) are in italics.
13. L505. "appear to wander (?) towards the opaque morphology" - use of "wander"
14. L630-632. "To the best of our knowledge, this is the first attempt to capture the space of *C. albicans*. morphology in a continuous manner that respects technical variation, developmental processes, and morphological transitions. " Not really true for the reasons mentioned above - work of Odds and colleagues.
15. L712713. Check subscripting - e.g. MgSO₄H₂O
16. Check for consistency - e.g. use e.g. of h, hrs and hours in the manuscript
17. Please use scale bars in all figures with micrographs.
18. K = kelvin, k = kilo
19. Some figures seem to have been stretched in production? E.g. Figure 3B (ii) - please check.

Reviewer #2 (Comments for the Author):

This paper introduces a system Candescence to detect and label the *C. albicans* cells in microscopic images. The authors propose a novel idea of applying a cumulative curriculum approach in deep learning. Then a Variational AutoEncoder is used to show the ubiquitous imperfect separation between all of the morphologies, signifying the complexity of *C. albicans* morphologies. Finally, a generative adversarial network is used to capture the biologically-relevant processes such as developmental trajectories and transition between morphologies. This model can also identify subtle changes in the phenotype due to genetically perturbed *C. albicans* populations. Good explanation of the cases where, when, how, and why Candescence does not perform well. This will help the community to improve the performance further in future by investigating those limitations.

Comments:

Use of the cumulative curriculum approach in developing the Candescence system is very interesting and seems useful based

on the experimental results and the discussion provided. The authors mentioned that they are the first to use this approach. But in line 264, saying that it has re-emerged in deep learning gives an impression that this paper is not the first or this concept of applying it in deep learning models is not novel. Perhaps authors can rephrase line 264 to emphasize that they are the first to apply it.

In the abstract, the authors just say their system gives very good performance but I wonder if they can also mention the obtained recall and precision. It will give a better idea how good it works.

I wonder if you tried to change the number of layers/neurons in ResNet-101 to see if it brings any significant improvements in your performance. Since you are doing transfer learning, maybe you could skip some layers or increase dropout probability to make the network smaller (in an easy way), then retrain, and see if it can raise the performance above 85%. Sometimes, too big (ResNet-101 has 4.5M parameters) network makes the optimization harder for the target problem. Sometimes people actually do network pruning to reduce the size of models while minimizing loss in accuracy or performance as well. You may give it a thought.

I wonder if it is possible to apply k-fold cross validation. Usually in deep learning or machine learning domains, people apply cross-validation to show that their model is not overfitting for a particular context or setting of the dataset.

Line 52: Why are the authors saying "large compendium of manually labelled images"? I see the number of manually labelled images is only about 1200.

Line 139: The authors say about a one-stage object detector but do not give any intuition about what is the specialty of this FCOS model? I wonder if they could very briefly mention here why they chose this model.

Line 157: It seems like this is a 9-category classification problem, where all those 9 categories represent some morphological state. I wonder why a category was not included that represents normal or background cells, or just artifacts. What does the model do when it sees something that does not belong to any of those 9 categories? Does the Softmax output layer distributes equal probabilities to all the classes then?

Line 174, when the authors first mention transfer learning, they should also mention why they need transfer learning. Usually we need transfer learning when our own target problem has a very small dataset that is not enough to train a deep learning model from scratch. In this paper, the target problem has a dataset of size about 1200 images, which is mentioned later in the paper. I think that size can be mentioned here so that the readers easily know why transfer learning is used.

Line 193 or 246, I wonder if authors could mention the number of cells and *C. albicans* cells in each image. This would reflect the difficulty of the problem easily.

Line 291: Is this validation step performed every training epoch to judge if the model is overfitting? This is not clearly mentioned. Based on the statements it seems like validation is a completely separate step to fine tune the model. In that case, what is the stopping criteria for the training? How do you know if the model is not overfitting during training? Because fine tuning an overfitted model that is already converged may not bring significant change because gradient does not update much after convergence.

Line 325: Does the ground truth set means the test set?

Line 327: It says that artifact is a heterogeneous class. Is this class included as a category among those 9-categories? This is not clear.

Line 354: It is saying 15 classes. But it is a 9-category classification problem. This is not clear why there are 15 classes in the confusion matrix.

Staff Comments:

Preparing Revision Guidelines

To submit your modified manuscript, log onto the eJP submission site at <https://spectrum.msubmit.net/cgi-bin/main.plex>. Go to Author Tasks and click the appropriate manuscript title to begin the revision process. The information that you entered when you

first submitted the paper will be displayed. Please update the information as necessary. Here are a few examples of required updates that authors must address:

Please return the manuscript within 60 days; if you cannot complete the modification within this time period, please contact me. If you do not wish to modify the manuscript and prefer to submit it to another journal, please notify me of your decision immediately so that the manuscript may be formally withdrawn from consideration by Microbiology Spectrum.

This paper introduces a system Candescence to detect and label the *C. albicans* cells in microscopic images. The authors propose a novel idea of applying a cumulative curriculum approach in deep learning. Then a Variational AutoEncoder is used to show the ubiquitous imperfect separation between all of the morphologies, signifying the complexity of *C. albicans* morphologies. Finally, a generative adversarial network is used to capture the biologically-relevant processes such as developmental trajectories and transition between morphologies. This model can also identify subtle changes in the phenotype due to genetically perturbed *C. albicans* populations. Good explanation of the cases where, when, how, and why Candescence does not perform well. This will help the community to improve the performance further in future by investigating those limitations.

Comments:

Use of the cumulative curriculum approach in developing the Candescence system is very interesting and seems useful based on the experimental results and the discussion provided. The authors mentioned that they are the first to use this approach. But in line 264, saying that it has re-emerged in deep learning gives an impression that this paper is not the first or this concept of applying it in deep learning models is not novel. Perhaps authors can rephrase line 264 to emphasize that they are the first to apply it.

In the abstract, the authors just say their system gives very good performance but I wonder if they can also mention the obtained recall and precision. It will give a better idea how good it works.

I wonder if you tried to change the number of layers/neurons in ResNet-101 to see if it brings any significant improvements in your performance. Since you are doing transfer learning, maybe you could skip some layers or increase dropout probability to make the network smaller (in an easy way), then retrain, and see if it can raise the performance above 85%. Sometimes, too big (ResNet-101 has 4.5M parameters) network makes the optimization harder for the target problem. Sometimes people actually do network pruning to reduce the size of models while minimizing loss in accuracy or performance as well. You may give it a thought.

I wonder if it is possible to apply k-fold cross validation. Usually in deep learning or machine learning domains, people apply cross-validation to show that their model is not overfitting for a particular context or setting of the dataset.

Line 52: Why are the authors saying “large compendium of manually labeled images”? I see the number of manually labeled images is only about 1200.

Line 139: The authors say about a one-stage object detector but do not give any intuition about what is the speciality of this FCOS model? I wonder if they could very briefly mention here why they chose this model.

Line 157: It seems like this is a 9-category classification problem, where all those 9 categories represent some morphological state. I wonder why a category was not included that represents normal or background cells, or just artifacts. What does the model do when it sees something that does not belong to any of those 9 categories? Does the Softmax output layer distributes equal probabilities to all the classes then?

Line 174, when the authors first mention transfer learning, they should also mention why they need transfer learning. Usually we need transfer learning when our own target problem has a very small dataset that is not enough to train a deep learning model from scratch. In this paper, the target problem has a dataset of size about 1200 images, which is mentioned later in the paper. I think that size can be mentioned here so that the readers easily know why transfer learning is used.

Line 193 or 246, I wonder if authors could mention the number of cells and *C. albicans* cells in each image. This would reflect the difficulty of the problem easily.

Line 291: Is this validation step performed every training epoch to judge if the model is overfitting? This is not clearly mentioned. Based on the statements it seems like validation is a completely separate step to fine tune the model. In that case, what is the stopping criteria for the training? How do you know if the model is not overfitting during training? Because fine tuning an overfitted model that is already converged may not bring significant change because gradient does not update much after convergence.

Line 325: Does the ground truth set means the test set?

Line 327: It says that artifact is a heterogeneous class. Is this class included as a category among those 9-categories? This is not clear.

Line 354: It is saying 15 classes. But it is a 9-category classification problem. This is not clear why there are 15 classes in the confusion matrix.

Reviewer #1:

“This is an interesting technical paper that describes a new image analysis tool -Candescence that can discriminate between none nine morphologies generated by the human pathogenic fungus *Candida albicans*. The papers is reasonably clearly written and the utility of the tool is demonstrated in the analysis of *Candida* mutant phenotypes.”

We thank the reviewer for a fantastic and detailed review. We learnt a lot from your comments and references to the literature. We hope that our changes in the document reflect your insight. We apologize for the lack of rigour with the literature and with some basic facts about morphology; *C. albicans* is a new area for the Hallett lab and our understanding remains far from complete.

There are some loosely, and informally written sections that could be sharpened and the paper misses some key foundation information in the literature that could and should be incorporated. The micrographs need space bars and the manuscript in general needs to be carefully checked for precision and consistency. It is a very long narrative and in my view this shortening would make it sharper and more accessible.

We address each of these shortcomings below in the line by line commentary. In general here, we have shortened the presentation of the manuscript. For example, we moved the section on the variational autoencoder (including Figure 4) to the supplemental as the paper remains understandable without this analysis.

We have also sharpened our language in places with clearer definitions especially with respect to deep learning concepts and measures of performance. We have placed space bars in the micrographs.

1. L 37-39 Abstract: " we then develop models using generative adversarial networks and identify subcomponents of the latent space which control technical variables" won't be understood my *Candida* biologists."? In general, if the specialists computational terminology could be explained carefully that would be helpful.

We agree and we have re-worked both the Abstract and Importance.

2. L53 If morphology index is a continuum how do you define break points? I see this is addressed later in the manuscript (L455-465). Important - it should be note dat this stage that this problem has been addressed previously by Odds et al (see below) who generated a very simple and useful morphology index.

We have modified the text to incorporate the prior work by Odds et al.

3. L74. Forms related to mating are rare in mucosal or skin surfaces.

Yes there was confusion in our presentation of the morphologies which are now fixed.

4. L78 I would not describe opaque cells as rectangular

Replaced with **The *opaque* form is larger and more elongated than spherical white cells.**

5. L96-98 presentation. "Pseudohyphae tend to have more branch points, because mother-daughter attachments are more easily disrupted than in hyphae..". I'm not sure this is the right interpretation.

Pseudohyphae tend to be synchronously dividing and that creates more evaginations that are seen in true hyphae where sub-spacial compartments are often in cell cycle arrested and do not re-enter the cell cycle for several generations after they were created. (i.e. the number of branches is not due to breakage as suggested here).

Hyphal branching frequency in hyphae is regulated by nutrient composition which affects how long a compartment remains arrested in the cell cycle,
(see

- Sudbery et al 2011, Growth of *Candida albicans* hyphae, *Nat Rev Microbiol*; PMID: 19347730;
- Sudbery, P., Gow, N.A.R. & Berman, J. (2004). The distinct morphogenic states of *Candida albicans*. *Trends in Microbiology* 12, 317-325.
- Veses, V & Gow N.A.R. (2009). Pseudohypha budding patterns of *Candida albicans*. *Medical Mycology* 47, 268-275.; and
- Barelle, C.J., Bohula, E.A., Kron, S.J., Wessels, D., Soll, D.R., Schäfer, A., Brown, A.J.P & Gow N.A.R. (2003). Asynchronous cell cycle in true hyphae of *Candida albicans* is related to asymmetric vacuolar inheritance. *Eukaryotic Cell* 2, 398-410.

Ok, thank you very much for your explanation. We have modified the text in the Introduction to incorporate these ideas with citations.

6. L103. Chlamydiae are very common on the appropriate media - but they are not usually seen in clinical samples.

I have removed the "less frequent" modifier from the sentence. It wasn't necessary for this manuscript.

7. L104. The work by Odds et al who devised a "morphology index" is perhaps the closest and most relevant early work to this publication, but it is not cited. See

- L A Merson-Davies 1, F C Odds. *J Gen Microbiol* . 1989 Nov;135(11):3143-52. doi: 10.1099/00221287-135-11-3143. A morphology index for characterization of cell shape in *Candida albicans*. PMID: 2693594; DOI: 10.1099/00221287-135-11-3143.
- L A Merson-Davies 1, V Hopwood, R Robert, A Marot-Leblond, J M Senet, F C Odds. 1991 Dec;35(6):321-4. *J Med Microbiol* doi: 10.1099/00222615-35-6-321. Reaction of *Candida albicans* cells of different morphology index with monoclonal antibodies specific for the hyphal form. PMID: 1753389; DOI: 10.1099/00222615-35-6-321.

Wow, those are great papers - and from 1989. I did not find these during my previous literature searches.

The present study was undertaken to investigate whether morphological forms of *C. albicans* could be quantified mathematically and rapidly by the use of computerized image analysis. Precise mathematical definition of morphologies should facilitate definition of cellular and molecular markers specific for different aspects of cell development in *C. albicans*. It would also allow determination of the distribution of defined morphological types that occur in different environmental conditions.

Yes, we entirely agree! It is a great motivation especially for the second part of our manuscript (use of the GAN to detect outliers). They suggest very simple models, and parameterizations, to capture the concept of morphology and subtle shifts between morphologies. It would be interesting to try and build a deep learner to parameterize these restrictive models. It would certainly be a challenge given how crowded and technically flawed some images are, but it could perhaps provide additional information with the existing Candescence discovery models. Also, it would be interesting to see if there is any evidence that deep learner is actually learning the types of ratios that they use in their models.

We have re-written the Introduction and relevant parts of the Discussion to correctly state the history of these approaches and problems (L 131-134) Thank you.

8. L190. Does this mean the total magnification or is this the objective magnification - the total magnification includes the objective mag and any further magnification of the image on the page.

This is only the objective magnification. We have made these issues clearer in the document.

9. Figure 1. Singular and plural of hypha/ hyphae/ pseudohypha/ Pseudohyphae e.g. the "the start of the pseudohyphae". e.g. Figure 5B(iii) Check throughout the manuscript.

We apologize for this. It is consistent now.

10. Have a space between the units and values - e.g. L 196. L214, L215 etc

It is consistent now.

11. The morphology of any growth form will also depend on the growth medium. It would be useful to explore the effect of nutrient status, temperature etc on the morphology of key cell types.

We fully agree. In this manuscript, the large Supplemental Table 1 describes the 61 different conditions we used as a far pass. The primary differences between these experiments is media and temperature, although ~10 represent genetically modified strains. This is far from a systematic exploration of how single variables or multiple variables concomitantly influence the distribution of observed morphologies. We are currently completing a second and third manuscript that explores this in a more detailed controlled setting. In the first followup, we have used Candescence on images from each strain of a large systematic deletion library. In the second followup, we have varied

temperature and media of yeast white cells grown in the presence of murine macrophages. In both cases, we see interesting shifts in the distribution of morphologies that - at least in some cases- can be traced back to molecular mechanisms.

12. L311. "Candescence almost never hallucinates but has some blind spots." Use of "hallucinates" is amusing, but may led to confusion. "generates false predictions" is better.

We agree and for false negatives (FNs), we can safely remove the more colorful blindspots. However, the dual nature of object recognition/object labelling problems such as ours here, there is some nuance that requires additional termination for false positives (FPs). On lines 772-778, we address this formally.

"An object is a FP if and only if either (i) the predicted bounding box does not overlap sufficiently with a ground truth bounding box, or (ii) the predicted bounding box does overlap sufficiently with a ground truth bounding box but the classification is incorrect. Type (i) is termed a *hallucination*. Type (ii) is a misclassification.

For clarity, type (i) means that Candescence predicts a bounding box in the middle of nowhere whereas type (ii) means that Candescence is correct to believe that there is a cell there but gets its morphology incorrect.

We have changed the poetic "hallucination" to "false object detection".

(For FNs it is simpler: FN if and only if we fail to predict a bounding box with sufficient overlap with a ground truth bounding box. The term FN can easily replace the more poetic *blindspots*.)

l387 - mutants (*brg1*) are in italics.

Fixed.

13. L505. "appear to wander (?) towards the opaque morphology" - use of "wander"

We have changed the language. (The machine/deep learning community oftens uses such floral language to help explain complicated mathematical and algorithmic concepts.)

14. L630-632. "To the best of our knowledge, this is the first attempt to capture the space of *C. albicans*. morphology in a continuous manner that respects technical variation, developmental processes, and morphological transitions. " Not really true for the reasons mentioned above - work of Odds and colleagues.

We have adjusted the text to include this important literature.

15. L712713. Check subscripting - e.g. $MgSO_4 \cdot 7H_2O$

Thank you.

16. Check for consistency - e.g. use e.g. of h, hrs and hours in the manuscript

Fixed.

17. Please use scale bars in all figures with micrographs.

Added.

18. K = kelvin, k = kilo

Fixed.

19. Some figures seem to have been stretched in production? E.g. Figure 3B (ii) - please check.

Fixed. Thank you.

Reviewer #2 (Comments for the Author):

This paper introduces a system Candescence to detect and label the *C. albicans* cells in microscopic images. The authors propose a novel idea of applying a cumulative curriculum approach in deep learning. Then a Variational AutoEncoder is used to show the ubiquitous imperfect separation between all of the morphologies, signifying the complexity of *C. albicans* morphologies. Finally, a generative adversarial network is used to capture the biologically-relevant processes such as developmental trajectories and transition between morphologies. This model can also identify subtle changes in the phenotype due to genetically perturbed *C. albicans* populations. Good explanation of the cases where, when, how, and why Candescence does not perform well. This will help the community to improve the performance further in future by investigating those limitations.

We thank you for your support, and your detailed insightful review below. We try to address each of your comments below.

Use of the cumulative curriculum approach in developing the Candescence system is very interesting and seems useful based on the experimental results and the discussion provided. The authors mentioned that they are the first to use this approach. But in line 264, saying that it has re-emerged in deep learning gives an impression that this paper is not the first or this concept of applying it in deep learning models is not novel. Perhaps authors can rephrase line 264 to emphasize that they are the first to apply it.

There are two concepts here. The first is the curriculum approach, which divides the learning set in such a way that the individual subsets of examples can be ordered by difficulty. Although the concept of curriculum learning has re-emerged in the deep learning community, it remains mostly unexplored (citations 56 and 57 essentially just introduce the notion of a curriculum and argue why this concept could in theory help in some learning situations). There is debate amongst practitioners about how to set up curriculum learning. The typical approach we found from the few examples in the literature is

to partition the learning set in say k grades. Training for grade i uses only examples from grade i , and proceeds from grade 1 to k .

We make a small contribution here with the second concept which is the idea of cumulative curriculum learning. Here the easiest examples (grade 1) are included in all grades 2-6. Generalizing, all examples in grade i are included in subsequent grades $i+1$, $i+2$, ... We hypothesize that this approach works better than non-cumulative curriculum learning because it penalizes the learner from “forgetting” easier examples. When we tried the first approach, the learner quite figured out the “easy” yeast white examples in grade 1, but the circuitry that codes for this capacity diverged after it was allowed to evolve with the complicated (hyphal) examples in grades 4-6. (Specifically, at the end of training, it lost its capacity to identify simple white cells well.) It is this “cumulative curriculum” learning which we believe is novel: we think the cumulative curriculum learning can replace in some cases the need for searching for optimal freezing strategies.

We have adjusted the language to make this clearer.

In the abstract, the authors just say their system gives very good performance but I wonder if they can also mention the obtained recall and precision. It will give a better idea how good it works.

We agree, and have added the recall and precision scores to the abstract.

I wonder if you tried to change the number of layers/neurons in ResNet-101 to see if it brings any significant improvements in your performance. Since you are doing transfer learning, maybe you could skip some layers or increase dropout probability to make the network smaller (in an easy way), then retrain, and see if it can raise the performance above 85%. Sometimes, too big (ResNet-101 has 4.5M parameters) network makes the optimization harder for the target problem. Sometimes people actually do network pruning to reduce the size of models while minimizing loss in accuracy or performance as well. You may give it a thought.

This is an interesting idea although there are problematic components if we were to allow ResNet-101 to be retrained or modified. Remember here that we are using transfer learning, so ResNet-101 is largely fixed and we are not learning the mentioned 4.5M parameters.

We do in fact experiment with allowing some of the highest levels of ResNet to be variable during retraining after the transfer. In MMDetection, this is controlled by the `frozen_layers` parameter. We experimented with different protocols for each grade of the curriculum learning. This is depicted in **Figure 1H** (freezing). It turns out that the best performance was achieved when only the first layer of ResNet-101 was allowed to vary (in fact, we had near similar performance when ResNet was not allowed to vary at all).

As per your intuition, this might be explained by the fact that the system has many parameters relative to our training data (ResNet does after all have 101 layers and was trained on orders of magnitude more data and very different data).

It of course remains an open question to optimize better and improve upon the ~85% performance. However, our analyses instead suggests that the classification errors occur between cells “on the border” between two morphologies (informally small opaque cells and large white cells). This is

supported by the confusion matrix in Figure 3C and the manual analysis of Figures 3A and B. It may be very difficult to resolve this issue. We can point to examples of a cell in a specific image that overwhelmingly contains white morphology cells, and one cell in another image that is predominantly of the opaque morphology, and such that these two cells are so close in appearance that an expert will not be able to decide. Perhaps better optimizations would result from designing networks that use contextual information for each cell. Informally, we would try encode conditional dependencies into the network of the form:

Pr[cell c is white | cell c is surrounded by 85% white cells and 15% opaque cells in the image]

We do agree that ResNet may not be the optimal pre-existing neural network to use in microscopy. To the best of our knowledge ResNet was not trained on microscopy images. The types of shading, edges, intersections and textures of DIC images seem qualitatively different than images of toasters, cats and road sides. We know of two efforts currently by large AI companies that are trying to train a very deep network only on microscopy images; it will be interesting to see the effect on the performance of Candescence.

I wonder if it is possible to apply k-fold cross validation. Usually in deep learning or machine learning domains, people apply cross-validation to show that their model is not overfitting for a particular context or setting of the dataset.

The reviewer is correct to focus on issues of over-fitting and over-learning; this was a constant concern for us during each step of the computation. Much of this data science was omitted from the manuscript due to (i) the nature of the audience, (ii) because our methods follow standard protocols including from the original FCOS papers, (iii) and all of our code and analyses are available as open source as part of the paper. Technically sophisticated deep learning people could easily repeat all of our analyses to test for this, if they were interested.

For clarity here before proceeding, recall that we are not optimizing on accuracy, precision, F1 etc. but rather using a deep learning approach via the FCOS that exploits gradient descent along the differentiable loss functions. There are three loss functions in an FCOS: (1) for the regression of object location e.g. top left and bottom right corner of box, (2) for so-called “centerness” that is a trick used by FCOS to identify a single bounding box focused properly on the object, and (3) the classification loss when determining morphology. Although there is some previous work to use k-fold cross validation in deep learning contexts, it is not standard in this field, there is some debate as to its efficacy and it has not been explored to date in the context of an FCOS. It is of course an interesting idea but there are some tricky issues here:

The standard set-up for the deep learning community (which we have used here) is to have a tri-partition of our learning dataset: the training, validation and test sets. The training dataset is fed to the network in minibatches, and the validation dataset is used to judge to update the network after each epoch along the differentials of the backpropagation algorithm. Neither validation nor datasets are used for training. The test dataset, which is completely independent from the learning phase, is used to evaluate the overall performance including measuring over-fitting. The main difficulty here with using k-fold CV is that the validation datasets would be very small and the optimization (backpropagation) might suffer as to negate any benefit towards ablating over-learning. Recall also that in our particular case here there are added complexities. First, the curriculum learning requires

that we partition the dataset by images, and each image may contain up to ~100 individual cells. It already is very tricky, requiring iteration and heuristics, to find first a tri-partition (train, validation, test) and then second to find a further 6-grade partition within each of the tri-partitions with various properties:

(i) the vast majority of cells in images that were assigned to grade i are simpler than the cells in images assigned to grades $>i$,

(ii) all grades have at least some examples of all of the 15 classes (a technical issue that seems to assist convergence), and

(iii) all grades in each of the tri-partitions are of roughly the correct size.

This would now have to be repeated k times.

We have used standard techniques for assessing when to halt the learning process; this is presented in Supplemental Figure 2. Here, we have generated data far beyond the inflection point in the plot for each grade 1 through 6. This is to show that our learners are stable. We stop learning using standard inflection point analyses. Unsurprisingly the stopping point is grade specific.

Although we did not go as far as to implement a k -CV, over the months of experimentation, we built classifiers many times on different randomizations of the learning sets. For example, we would identify problems and inconsistencies in Varasana, fix them, and then re-design the tri-partition. The overall performance of the classifiers improved in lock-step with these improvements. (In fact, by mistake once, we included the training set in the validation dataset and our performance was basically perfect, and served as a strange type of positive control for over-fitting.)

Of course, it is correct to point out that some over-fitting/over-learning is subtle and cannot be captured or identified by any of these methods. Sometimes it simply takes multiple datasets from independent groups to truly assess whether any one particular learning set was biased in a certain direction. We cannot rule this out. We are nearing completion of two additional manuscripts that used completely independent datasets and the performance of Candescence was largely comparable. These two new datasets analyze genetically modified cells so it is difficult to judge whether the minor decrease in performance is due to true differences in morphology caused by the molecular challenges, or due to overfitting from Varasana. We are confident though that the over-fitting is not severe in our case here.

Line 52: Why are the authors saying "large compendium of manually labelled images"? I see the number of manually labelled images is only about 1200.

We have adjusted the language to remove the qualitative term. We note however that the 1200 images each contain on average ~25 cells (or 40 objects across all 15 classes) and each object needs to be carefully bounded by hand and then labelled with a morphology (using Labelbox). At nearly 48,000 objects, this represented a significant amount of time: although many cells are easy to label in terms of their morphology, other individual cells required >10 minutes of group discussion. As I am sure you understand, the creation of the correct ontology itself for labelling (objective criteria for determining an artifact vs. unknown vs. a specific morphology), coping with outliers and deciding on rules for morphology across many exceptional cases requires iteration over the dataset multiple times. Albeit imperfect, the authors do feel that the Varasana dataset represents a significant resource for the community and it is sufficiently "large" as to make it

arduous for others to create a similar dataset. Like our two new efforts above, Varasana serves as a starting point via transfer learning for new third-party datasets to be trained.

Line 139: The authors say about a one-stage object detector but do not give any intuition about what is the specialty of this FCOS model? I wonder if they could very briefly mention here why they chose this model.

We are somewhat concerned that we are misunderstanding the reviewer's question since the first subsection of the Results did contain the following:

162 object detector (FCOS)⁵⁵, a new approach that has several benefits over other deep learning
163 algorithms for multi-object, multi-class problems (**Figure 1F**). One of its key advantages resides
164 from how it flags areas of an image likely to contain an object. Rather than requiring many
165 parameters, which collectively control the size, location and total number of bounding boxes,
166 the FCOS considers each individual pixel in an image as potentially centering an object, ablating
167 the need to optimize many hyperparameters simultaneously. FCOSs are able to handle objects
168 of variable size, an important property given the difference between, for example, yeast white
169 and hyphal cells.

However, we have added sentences to the Introduction now to explain our rationale for an FCOS over say Faster R-CNN.

Line 157: It seems like this is a 9-category classification problem, where all those 9 categories represent some morphological state. I wonder why a category was not included that represents normal or background cells, or just artifacts. What does the model do when it sees something that does not belong to any of those 9 categories? Does the Softmax output layer distributes equal probabilities to all the classes then?

Figure 1A-E depict the different types of cells. To clarify here, there is no such thing as a "normal" *C. albicans* cell per se. All cells have a morphology and that morphology depends on the context in which it is grown. Perhaps white cells are the closest to a "normal" cell and they are our first class (Figure 1B).

Also in Figure 1, we introduce two additional technical classes termed "Unknown" and "Artifact". These are explained on Lines 220 and 221:

"Those cells for which we could not reach agreement on morphology were labelled as *unknown* and non-cellular events in the images were labelled as *artifacts*."

There were a few (<5) very strange looking cells which were also labelled as "unknown", but they are likely still *C. albicans* cells.

We have tried to re-inforce that the first Results subsection is structured to begin with the biological goal (which is to predict 9 of the 12 morphologies) and to describe the technical details below.

It turns out that there was no evidence that Softmax outputs equal probabilities across all 15 classes when the cell does not look like one of the 9 morphologies. We discuss this further below. It turns out that Candescence can pigeonhole very strange looking objects as either “artifacts” or “unknowns” with reasonable accuracy (so the probabilities are centered on one or both of these groups). We too anticipated larger problems with this issue but it turned out to not be a problem. See discussion below.

Line 174, when the authors first mention transfer learning, they should also mention why they need transfer learning. Usually we need transfer learning when our own target problem has a very small dataset that is not enough to train a deep learning model from scratch. In this paper, the target problem has a dataset of size about 1200 images, which is mentioned later in the paper. I think that size can be mentioned here so that the readers easily know why transfer learning is used.

We have added a sentence that better motivates the use of transfer learning but note in general that transfer learning can be used in many contexts including when we are “data poor”. However, we politely remind the reviewer that we have 1200 images but there are up to 100 cells per image and there are close to 50,000 cellular objects. It is the latter number that is pertinent to training. Supplemental Table 1 describes these 1200 images and Supplemental Figure 1 provide the distributions of the number of cells per image.

Line 327: It says that artifact is a heterogeneous class. Is this class included as a category among those 9-categories? This is not clear.

Perhaps this was just an oversight, or perhaps we do not understand your question properly. It is stated in the first subsection of the results, in Figure 1, in the confusion matrix of Figure 3C, in Figure 4, Supplemental Figure 1 and elsewhere. Perhaps you mean something else. We explore this a bit here. It is not considered one of the 9 morphologies but it is one of the 15 classes of the deep learner.

The artifact class is heterogeneous because it includes any sort of technical artifact: scratches of microscopy slides, dust, dead cells, contaminants, etc. The point is that these different sources of technical artifacts do not share visual and structural similarities. Some look like black streaks and others look like white smears. Unknown is a slightly different class. Here are some examples of artifacts:

Objects labeled as unknown are cases where it is clearly a cell but for which we can't accurately assign a morphology.

There are at least three ways to deal with artifacts (and unknowns). The first approach is to remove the area of the bounding boxes for artifacts from the images entirely. Then they will never be learnt by the system. This is a very pure approach but this causes many technical problems. For instance, the FCOS is no longer using a rectangular image and "zero'd out" regions cannot be handled with the dynamic anchor search algorithm. It would require substantial effort to extend the underlying FCOS' technique. Moreover, the bounding box of an artifact might overlap with other objects and the removal of artifact bounding boxes will affect the capacity of the FCOS to identify and label these neighbors.

The second approach is not to label the objects. This causes many "false positives" after training because Candescence will identify and be "forced" to guess a morphology. This was tried and the error rates were substantial

The third approach is to label objects as artifacts and include the artifact class in the training. Ultimately, we do not care if the system predicts a lot of artifacts, as long as it doesn't predict artifacts where there are actually good cells. Also, we do not care if it misses some artifacts, as long as it doesn't predict cells where there are artifacts. It turns out that Candescence has neither problem as observed in the confusion matrix of Figure 3C. In fact, it is very good at identifying artifacts and unknowns. The mistakes for these two classes are mostly between artifacts and unknowns! Only 1 artifact was predicted as a shmoo!

So it turns out that using the simple solution solves the problem very nicely at least to date. It is true that Candescence misses many artifacts, and it also identifies many artifacts that were not labelled by the humans. The latter is because there are arguably hundreds of tiny artifacts in every image and humans are very lazy.

Line 193 or 246, I wonder if authors could mention the number of cells and *C. albicans* cells in each image. This would reflect the difficulty of the problem easily.

Especially in filamentous images, it is difficult often to determine exactly where one hypha starts and ends. To this end, Supplemental Figure 1 was included to provide the *distribution* of cells per image across the six grades of the curriculum with Supplemental Figure 1B providing the number of cells per image. The number of cells per image depends on the specific growth conditions and the genetic background of the cells. Figure 2 was included to address this issue of "complexity" with filamentous cases.

Line 291: Is this validation step performed every training epoch to judge if the model is overfitting? This is not clearly mentioned. Based on the statements it seems like validation is a completely separate step to fine tune the model. In that case, what is the stopping criteria for the training? How do you know if the model is not overfitting during training? Because fine tuning an overfitted model that is already converged may not bring significant change because gradient does not update much after convergence.

We are not sure of the source of confusion that led you to believe that we were fine tuning the classifier during the test step. To be clear: we do not, in any way, shape or form. We use standard deep learning approaches for training, assessing over-fitting and stoppage criteria. We do not jiggle the network in any way afterwards. We follow standard protocol for an FCOS and how it has been applied elsewhere. There are three datasets: training and validation - they are used in completely standard ways for training any convolutional network regardless of an FCOS or otherwise. The test set is completely independent and it is used to assess the capacity of the system including generalization/over-learning after training is complete and the network weights are “written in stone”.

Line 325: Does the ground truth set means the test set?

In this case in line 325 it refers to the validation set. This is a negative result and it says that even when applied to data that was used during the training-validation cycle (and therefore has a huge advantage in terms of over-learning) it still misses these. We go on to explain by “lifting the hood” with these examples and find that they are mostly related to artifacts. Since artifacts come in all shapes and sizes, it seems difficult to find them all.

Informally, the bottomline is that we don't care about artifacts. All microscopy images have them and they are a nuisance. In fact, humans perform very poorly at labelling *all* artifacts (several hundred per slide arguably). We just don't really care that Candescence misses these.

We have adjusted the text to be clearer.

Line 354: It is saying 15 classes. But it is a 9-category classification problem. This is not clear why there are 15 classes in the confusion matrix.

We consider 9 morphologies; these are described in the Introduction and then again more precisely in the first subsection of the Results section. We explain why the hyphal cells need additional classes using several lines (L 206 to 219). This was supported by Figure 1D and Figure 1E where we introduce the concept of the “start” and “junction” classes in addition to the “overall” hypha and pseudo-hypha classes that bound the entire object

3 x 2 (for hypha/pseudo) =6

As mentioned in the same subsection and depicted in Figure 1A, we also have two technical classes called Unknown and Artifact discussed above.

2 (artifact, unknown)

and the 7 remaining morphologies that have one class each

7 remaining morphologies = 15 classes for 9 morphologies

This was the intention of structuring the subsection of the results: we start with the biology (9 morphologies) and then give the details of the 15 classes below. We have tried to adjust the subsection again to make this more straightforward.

(What is really amazing to us is that the deep learner seems to have figured out that a p-start and p-junction occur near each other and within a pseudo-bounding box. This is also true for the h-start, h-junction and hypha-bounding box. At least the human eye has difficulty distinguishing

between p and h start/junctions, but Candescence doesn't have that confusion. We believe that it may be learning a rule that disallows predictions of, for example, p-start and h-junctions in the same object.)

Staff Comments:

Preparing Revision Guidelines

For complete guidelines on revision requirements, please see the journal Submission and Review Process requirements at <https://journals.asm.org/journal/Spectrum/submission-review-process>.

Submissions of a paper that does not conform to Microbiology Spectrum guidelines will delay acceptance of your manuscript. "

Completed.

July 25, 2022

Dr. Michael T Hallett
University of Western Ontario
Biochemistry
Medical Science Building M359
London, Ontario N6G 2V4
Canada

Re: Spectrum01472-22R1 (A deep learning approach to capture the essence of *Candida albicans* morphologies)

Dear Dr. Michael T Hallett:

I am pleased to inform you that your manuscript has been accepted for publication, and I am forwarding it to the ASM Journals Department for publication. You will be notified when your proofs are ready to be viewed.

Sincerely,

Rebecca Shapiro
Editor, Microbiology Spectrum

Journals Department
Supplemental Material 7: Accept
Supplemental Material 5: Accept
Supplemental Material 10: Accept
Supplemental Material 2: Accept
Supplemental Material 1: Accept
Supplemental Material 3: Accept
Supplemental Material 9: Accept
Supplemental Material 6: Accept
Supplemental Material 4: Accept
Supplemental Material 8: Accept